# An Information-theoretic Approach to Distribution Shifts

**Marco Federici**
AMLab
University of Amsterdam
`m.federici@uva.nl`

**Ryota Tomioka**
Microsoft Research
Cambridge, UK
`ryoto@microsoft.com`

**Patrick Forré**
AI4Science Lab, AMLab
University of Amsterdam
`p.d.forre@uva.nl`

## Abstract

Safely deploying machine learning models to the real world is often a challenging process. Models trained with data obtained from a specific geographic location tend to fail when queried with data obtained elsewhere, agents trained in a simulation can struggle to adapt when deployed in the real world or novel environments, and neural networks that are fit to a subset of the population might carry some selection bias into their decision process. In this work, we describe the problem of data shift from a novel information-theoretic perspective by (i) identifying and describing the different sources of error, (ii) comparing some of the most promising objectives explored in the recent domain generalization and fair classification literature. From our theoretical analysis and empirical evaluation, we conclude that the model selection procedure needs to be guided by careful considerations regarding the observed data, the factors used for correction, and the structure of the data-generating process.

## 1 Introduction

One of the most common assumptions for machine learning models is that the training and test data are independently and identically sampled (IID) from the same distribution. In practice, this assumption does not hold in many practical scenarios (Bengio et al., 2020). A machine learning model trained to recognize land usage from satellite images using pictures from the early 2000s may struggle to recognize the style of modern architectures (Christie et al., 2018), data collected on a limited set of hospitals may not be representative of the variation introduced by the use of different machines or procedures (Zech et al., 2018; Beede et al., 2020). Other kinds of distribution shifts are more subtle and difficult to recognize despite having a noticeable impact on the model's predictive performance. Examples include under-represented or over-represented population groups (Popejoy & Fullerton, 2016; Buolamwini & Gebru, 2018) or biased annotations collected from crowd-sourcing services (Zhang et al., 2017; Xia et al., 2020).

Different approaches in literature address these issues by using some external source of knowledge such as domain or environment annotations (Wang & Deng, 2018), protected attributes (Mehrabi et al., 2019) or sub-population groups (Santurkar et al., 2020) to reduce bias and minimize the model error outside of the training distribution. Despite the progress in the field of domain generalization literature, Gulrajani & Lopez-Paz (2020) has shown that the effectiveness of some of the most common algorithms heavily relies on the hyper-parameter tuning strategy, revealing limitations of

35th Conference on Neural Information Processing Systems (NeurIPS 2021).

examined models when compared to a more traditional empirical risk minimization strategy. One of the major issues behind this observed behavior is a lack of clarity and applicability for the underlying assumptions regarding the problem statement and the data-generating process (Zhao et al., 2019; Rosenfeld et al., 2020; Mahajan et al., 2020).

With this work, we aim to present a new perspective on the problem to analyze and clarify the fundamental differences between some of the most common approaches by:

1. Introducing a novel information-theoretical framework to describe the problem of distribution shift and connecting it to the test error and its components (section 2).

2. Analyzing four main families of objectives and describing some of their guarantees and assumptions (section 3).

3. Demonstrating that the effectiveness of different criteria is determined by the structure of the underlying data-generating process (section 4).

4. Showing that the results obtained by popular models designed according to the aforementioned criteria can drastically differ from the theoretically expected performance (section 4).

The analysis of the *bottleneck*, *independence*, *sufficiency*, and *separation* criteria reveals that some of the most popular models can systematically fail to reduce the test error even for simple datasets. No unique objective is simultaneously optimal for every problem, but additional knowledge about the selection procedure and better approximations can help to mitigate the bias.

## 1.1 Problem Statement

Consider $\mathbf{x}$ and $\mathbf{y}$ as the features and targets respectively with joint density $p(\mathbf{x}, \mathbf{y})$ for the predictive problem of interest. Let t be a binary random variable representing which data is selected for training (t = 1) and which is not (t = 0). We will refer to $p(\mathbf{x}, \mathbf{y}|t = 1)$ as the joint *Training distribution* and $p(\mathbf{x}, \mathbf{y}|t = 0)$ as the *Test distribution* that are induced by the selection t. Throughout the paper, we will consider selections t that can be expressed as a function of the features $\mathbf{x}$, targets $\mathbf{y}$, independent noise $\epsilon$ and other variables $\mathbf{e}$, which represent other factors that are affecting the data collection procedure (e.g. geographic location, time intervals, population groups).

Let $q(\mathbf{y}|\mathbf{x})$ represent a learnable model for the predictive distribution $p(\mathbf{y}|\mathbf{x})$. We use the Kullback-Leibler divergence to express the *Train* and *Test* error respectively[1]:

$$\underbrace{D_{\mathrm{KL}}(p_{\mathbf{y}|\mathbf{x}}^{\mathrm{t=1}}||q_{\mathbf{y}|\mathbf{x}})}_{\text{Train error}} := D_{\mathrm{KL}}(p(\mathbf{y}|\mathbf{x}, \mathrm{t} = 1)||q(\mathbf{y}|\mathbf{x})), \tag{1}$$

$$\underbrace{D_{\mathrm{KL}}(p_{\mathbf{y}|\mathbf{x}}^{\mathrm{t=0}}||q_{\mathbf{y}|\mathbf{x}})}_{\text{Test error}} := D_{\mathrm{KL}}(p(\mathbf{y}|\mathbf{x}, \mathrm{t} = 0)||q(\mathbf{y}|\mathbf{x})). \tag{2}$$

Although only the train distribution can be accessed at training time, we are interested in learning models $q(\mathbf{y}|\mathbf{x})$ that result in small test error.

## 1.2 Characterizing Distribution Shift

As a first step, we characterize the difference between training and test distribution as a function of the selection. The effect that the selection has in the joint distribution $p(\mathbf{x}, \mathbf{y})$ can be quantified by considering the mutual information between the features-target pair $\mathbf{xy}$ and the selection variable t, which can be also expressed as a Kullback-Leibler divergence: $I(\mathbf{xy}; \mathrm{t}) = D_{\mathrm{KL}}(p(\mathbf{x}, \mathbf{y}|\mathrm{t})|p(\mathbf{x}, \mathbf{y}))$. This measure of distribution shift intuitively represents how many bits the selection variable carries about the joint distribution of targets and features or, equivalently, how much the joint density $p(\mathbf{x}, \mathbf{y})$ has changed as a result of the selection.

Using the chain rule of mutual information, one can express distribution shift as the sum of two separate components:

$$\underbrace{I(\mathbf{xy}; \mathrm{t})}_{\text{Distribution shift}} = \underbrace{I(\mathbf{x}; \mathrm{t})}_{\text{Covariate shift}} + \underbrace{I(\mathbf{y}; \mathrm{t}|\mathbf{x})}_{\text{Concept shift}}, \tag{3}$$

---

[1]Further details regarding the convention used for conditional KL-divergence can be found in appendix A

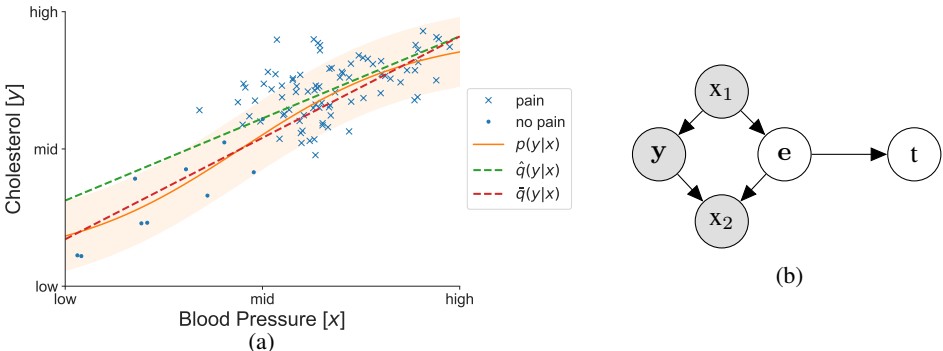

Figure 1: (a) Example of distribution shift due to selection bias (chest pain complaints) for the prediction of cholesterol levels (y-axis) given blood pressure (x-axis). A maximum likelihood approach (green dashed line) results in a model that over-estimates y for any given x when compared to the best model in the class (red dashed line). (b) Example of a data generating process in which the features $\mathbf{x}$ are composed of two parts $x_1$ and $x_2$. When both feature components $x_1$ and $x_2$ are observed, concept shift can be strictly positive. Discarding information about $x_2$ will reduce the effect of the selection bias, while removing $x_1$ might increase it $0 = I(\mathbf{y}; \mathbf{t}|x_1) \leq I(\mathbf{y}; \mathbf{t}|x_1 x_2) \leq I(\mathbf{y}; \mathbf{t}|x_2)$.

which can be interpreted as the amount of *covariate shift* (Shimodaira, 2000) and *concept shift* (Widmer & Kubat, 1996; Moreno-Torres et al., 2012), respectively. These two quantities refer to the changes in the predictive distribution $p(\mathbf{y}|\mathbf{x})$ and the marginal features distribution $p(\mathbf{x})$ respectively, which add up to represent the changes in the joint distribution.

Whenever the data selection is perfectly IID, the selection variable can be expressed as a function of some independent noise ($\mathbf{t} = f(\epsilon)$) and the corresponding distribution shift $I(\mathbf{xy}; \mathbf{t})$ is zero. On the other hand, if the data collection procedure has been influenced by other factors, we do not have such a guarantee, even when the selection does not depend on features and targets directly ($\mathbf{t} = f(\mathbf{e}, \epsilon)$).

## 2 Information-theoretic Framework

Using the quantities described in the previous sections, we can show that the sum train and test error is lower-bounded by the amount concept shift:

**Proposition 1.** *For any model $q(\mathbf{y}|\mathbf{x})$ and $\alpha := \min\{p(\mathbf{t}=0), p(\mathbf{t}=1)\}$:*

$$D_{\text{KL}}(p_{\mathbf{y}|\mathbf{x}}^{\mathbf{t}=1}||q_{\mathbf{y}|\mathbf{x}}) + D_{\text{KL}}(p_{\mathbf{y}|\mathbf{x}}^{\mathbf{t}=0}||q_{\mathbf{y}|\mathbf{x}}) \geq \frac{1}{1-\alpha}I(\mathbf{y}; \mathbf{t}|\mathbf{x}), \tag{4}$$

As a consequence, whenever the selection induces concept shift $I(\mathbf{y}; \mathbf{t}|\mathbf{x}) > 0$, any sufficiently flexible model $\hat{q}(\mathbf{y}|\mathbf{x})$ trained using a maximum likelihood approach must incur in strictly positive test error. Intuitively, whenever the selection induces a change in the predictive distribution ($p(\mathbf{y}|\mathbf{x}) \neq p(\mathbf{y}|\mathbf{x}, \mathbf{t}=1)$) fitting the model to the training distribution will incorporate the selection bias into the model prediction, necessarily resulting in errors when evaluated on the test distribution, as shown in the example reported in figure 1a.

The different approaches analyzed in this work are based in the introduction of a *latent representation* $\mathbf{z}$, which allows for the definition of different regularization strategies. Using the latent representation as an intermediate variable, the model $q(\mathbf{y}|\mathbf{x})$ can be re-parametrized with an *encoder* $q(\mathbf{z}|\mathbf{x})$ and a *latent classifier*[2] $q(\mathbf{y}|\mathbf{z})$:

$$q(\mathbf{y}|\mathbf{x}) = \mathbb{E}_{\mathbf{z} \sim q(\mathbf{z}|\mathbf{x})}\left[q(\mathbf{y}|\mathbf{z}=\mathbf{z})\right]. \tag{5}$$

The effect of the introduction of a latent representation can be observed by expressing the training and test error as a function of the encoder and the classifier[3].

---

[2]The name classifier will be used for convenience to express both classification and regression problems.

[3]The expressions in equations 6 and 7 hold with equality for deterministic encoders as shown in appendix B.

**Proposition 2.** *For any encoder $q(\mathbf{z}|\mathbf{x})$ and classifier $q(\mathbf{y}|\mathbf{z})$:*

$$D_{\mathrm{KL}}(p_{\mathbf{y}|\mathbf{x}}^{\mathrm{t=1}}||q_{\mathbf{y}|\mathbf{x}}) \leq \underbrace{I_{t=1}(\mathbf{x};\mathbf{y}|\mathbf{z})}_{\text{Train information loss}} + \underbrace{D_{\mathrm{KL}}(p_{\mathbf{y}|\mathbf{z}}^{\mathrm{t=1}}||q_{\mathbf{y}|\mathbf{z}})}_{\text{Latent train error}} \tag{6}$$

$$D_{\mathrm{KL}}(p_{\mathbf{y}|\mathbf{x}}^{\mathrm{t=0}}||q_{\mathbf{y}|\mathbf{x}}) \leq \underbrace{I_{t=0}(\mathbf{x};\mathbf{y}|\mathbf{z})}_{\text{Test information loss}} + \underbrace{D_{\mathrm{KL}}(p_{\mathbf{y}|\mathbf{z}}^{\mathrm{t=0}}||q_{\mathbf{y}|\mathbf{z}})}_{\text{Latent test error}}. \tag{7}$$

The two terms $I_{t=1}(\mathbf{x};\mathbf{y}|\mathbf{z})$ and $I_{t=0}(\mathbf{x};\mathbf{y}|\mathbf{z})$ represent the amount of predictive information that is lost by encoding the features $\mathbf{x}$ into $\mathbf{z}$ on train and test distribution respectively, while $D_{\mathrm{KL}}(p_{\mathbf{y}|\mathbf{z}}^{\mathrm{t=1}}||q_{\mathbf{y}|\mathbf{z}})$ and $D_{\mathrm{KL}}(p_{\mathbf{y}|\mathbf{z}}^{\mathrm{t=0}}||q_{\mathbf{y}|\mathbf{z}})$ refer to the train and test error when using $\mathbf{z}$ instead of the original observations as the predictive features, which will be referred to as *latent training error* and *latent test error* respectively. Test information loss and latent test error capture two intrinsically different kinds of error. The former indicates the increase in the prediction uncertainty as a result of the encoding procedure, while the latter represents the discrepancy between the model $q(\mathbf{y}|\mathbf{z})$ and the latent test predictive distribution $p(\mathbf{y}|\mathbf{z},\mathrm{t}=0)$.

## 2.1 Latent test error

Minimizing the latent test error makes the latent predictor $q(\mathbf{y}|\mathbf{z})$ approach the test latent predictive distribution $p(\mathbf{y}|\mathbf{z},\mathrm{t}=0)$. We can show that the Jensen-Shannon divergence between the two distributions is upper-bounded by a monotonic function of the latent training error and the amount of concept shift in the latent space (*latent concept shift $I(\mathbf{y};\mathrm{t}|\mathbf{z})$*):

**Proposition 3.** *For any $q(\mathbf{z}|\mathbf{x})$, $q(\mathbf{y}|\mathbf{z})$ and any representation $\mathbf{z}$ that satisfies $p(\mathbf{z}=\mathbf{z}|\mathrm{t}=0)>0$ and $p(\mathbf{z}=\mathbf{z}|\mathrm{t}=1)>0$:*

$$\left(\sqrt{\frac{1}{2\alpha}I(\mathbf{y};\mathrm{t}|\mathbf{z}=\mathbf{z})} + \sqrt{\frac{1}{2}D_{\mathrm{KL}}(p_{\mathbf{y}|\mathbf{z}=\mathbf{z}}^{\mathrm{t=1}}||q_{\mathbf{y}|\mathbf{z}=\mathbf{z}})}\right)^2 \geq D_{\mathrm{JSD}}(p_{\mathbf{y}|\mathbf{z}=\mathbf{z}}^{\mathrm{t=0}}||q_{\mathbf{y}|\mathbf{z}=\mathbf{z}}). \tag{8}$$

Whenever the Jensen-Shannon divergence between the test predictive distribution $p(\mathbf{y}|\mathbf{z},\mathrm{t}=0)$ and the classifier $q(\mathbf{y}|\mathbf{z})$ is small, the latent test error (measured in terms of KL-divergence) must also be small at least for the regions that have positive probability according to both train $p(\mathbf{x}|\mathrm{t}=1)$ and test $p(\mathbf{x}|\mathrm{t}=0)$ data distributions. Since the train predictive distribution $p(\mathbf{y}|\mathbf{x},\mathrm{t}=1)$ is not defined for $x$ that have zero probability on the train distribution, we have no guarantees regarding the model predictions in those regions unless other inductive biases are considered.

The left hand side of the expression in proposition 3 can be minimized by addressing $I(\mathbf{y};\mathrm{t}|\mathbf{z})$ and $D_{\mathrm{KL}}(p_{\mathbf{y}|\mathbf{z}}^{\mathrm{t=1}}||q_{\mathbf{y}|\mathbf{z}})$ with respect to the encoder and classifier respectively. In other words, we can minimize the latent test error by simultaneously fitting $q(\mathbf{y}|\mathbf{z})$ to the train predictive distribution $p(\mathbf{y}|\mathbf{z},\mathrm{t}=1)$ and minimizing the latent concept shift induced by the encoder $q(\mathbf{z}|\mathbf{x})$. When the latent concept shift and latent test error approach zero, the latent train predictive distribution $p(\mathbf{y}|\mathbf{z},\mathrm{t}=1)$ approaches the true (unselected) predictive distribution $p(\mathbf{y}|\mathbf{z})$, and so does the modeled classifier $q(\mathbf{y}|\mathbf{z})$. In this ideal scenario, the only source of test error is the additional uncertainty that is due to the information lost in the encoding procedure.

## 2.2 Minimizing the information loss

Since losing information generally results in increased test error (equation 7), we will consider objectives that discard the minimal amount of information required to reduce the latent concept shift. This can be done by minimizing the KL-divergence between the training predictive distribution $p(\mathbf{y}|\mathbf{x};\mathrm{t}=1)$ and the latent classifier model $q(\mathbf{y}|\mathbf{z})$:

$$\min_{q(\mathbf{z}|\mathbf{x})} I_{t=1}(\mathbf{x};\mathbf{y}|\mathbf{z}) = \min_{q(\mathbf{z}|\mathbf{x}),q(\mathbf{y}|\mathbf{z})} D_{\mathrm{KL}}(p_{\mathbf{y}|\mathbf{x}}^{\mathrm{t=1}}||q_{\mathbf{y}|\mathbf{z}}) - KL(p_{\mathbf{y}|\mathbf{z}}^{\mathrm{t=1}}||q_{\mathbf{y}|\mathbf{z}})$$

$$\leq \min_{q(\mathbf{z}|\mathbf{x}),q(\mathbf{y}|\mathbf{z})} D_{\mathrm{KL}}(p_{\mathbf{y}|\mathbf{x}}^{\mathrm{t=1}}||q_{\mathbf{y}|\mathbf{z}}). \tag{9}$$

In addition to reducing the amount of information lost in the encoding procedure, minimizing the right hand side of the expression in equation 9 makes the model $q(\mathbf{y}|\mathbf{z})$ approach the latent predictive

distribution $p(\mathbf{y}|\mathbf{z}, \mathbf{t} = 1)$. Note that only the selected training distribution $p(\mathbf{y}, \mathbf{x}|\mathbf{t} = 1)$ is available at training time. In practice, minimizing the train information loss also generally results in decreased test information loss since the same features are usually informative for both train and test distributions.

## 2.3 A General Loss function

To summarize, the overall objective consists in finding the maximally informative representation that minimizes latent concept shift so that the same learned predictor $q(\mathbf{y}|\mathbf{z})$ can perform similarly on both train and test settings. This is achieved by (i) minimizing the amount of latent concept shift induced by the representation, (ii) maximizing the amount of predictive information in the representation, and (iii) matching $q(\mathbf{y}|\mathbf{z})$ and the training latent predictive distribution $p(\mathbf{y}|\mathbf{z}, \mathbf{t} = 1)$. The three requirements can be enforced by considering a loss function with the following form:

$$\mathcal{L}(q_{\mathbf{z}|\mathbf{x}}, q_{\mathbf{y}|\mathbf{z}}; \lambda) = D_{\mathrm{KL}}(p_{\mathbf{y}|\mathbf{x}}^{\mathbf{t}=1}||q_{\mathbf{y}|\mathbf{z}}) + \lambda \mathcal{R}(q_{\mathbf{z}|\mathbf{x}}), \tag{10}$$

in which, the first term addresses (ii) and (iii), while the second term represent a regularization term that acts on the encoder $q(\mathbf{z}|\mathbf{x})$ to minimize latent concept shift $I(\mathbf{y}; \mathbf{t}|\mathbf{z})$, following (i). For a sufficiently flexible family of latent predictors, requirement (iii) depends only $q(\mathbf{y}|\mathbf{z})$, while the hyper-parameter $\lambda$ defines the trade-off between latent concept shift (i) and predictive information loss (ii).

## 3 Regularization Criteria

Since only selected data ($\mathbf{t} = 1$) is accessible at training time, the latent concept shift can not be computed or minimized directly. Most of the approaches considered in this analysis make use of a regularization $\mathcal{R}(q_{\mathbf{z}|\mathbf{x}})$ that is based on the observation of an additional variable $\mathbf{e}$ which relates to the selection criteria. This variable is usually referred to as *domain* or *environment*[4] in the domain adaptation and generalization literature, while the name *protected attribute* is used in the context of fair classification. We will refer to this variable $\mathbf{e}$ as *environmental factors* in the following sections.

We analyze four families of criteria proposed in the representation learning (Tishby & Zaslavsky, 2015), domain generalization (Koyama & Yamaguchi, 2020) and fair classification (Barocas et al., 2018) literature focusing on their underlying assumptions and theoretical guarantees. The different regularization strategies and models can be seen as specific instance of the loss function in equation 10. An empirical comparison between instances of the different criteria can be found in section 4, proofs are reported in appendix B, while the relation between the reported criteria is further discussed in appendix B.9.

### 3.1 Information Bottleneck Criterion

Combining the results from propositions 1 and 2 we can infer that reducing the latent test error necessarily requires the representation to discard some train predictive information ($I_{\mathbf{t}=1}(\mathbf{x}; \mathbf{y}|\mathbf{z}) > 0$). This is because a lossless encoder $\hat{q}(\mathbf{z}|\mathbf{x})$ together with an optimal latent classifier $\hat{q}(\mathbf{y}|\mathbf{z})$ would result in an overall model $\hat{q}(\mathbf{y}|\mathbf{x})$ that matches the train distribution $p(\mathbf{y}|\mathbf{x}, \mathbf{t} = 1)$, and, therefore, results in positive latent test error (proposition 1).

The *Information Bottleneck* criterion (Tishby & Zaslavsky, 2015) introduces a regularization term $\mathcal{R}(q_{\mathbf{z}|\mathbf{x}}) = I_{t=1}(\mathbf{x}; \mathbf{z})$ to control the amount of information in the representation, defining a lossy compression scheme (Alemi et al., 2017) that can be regulated using the hyper-parameter $\lambda$. Note that this criterion does not use any environmental information and blindly discards data-features depending on the regularization strength $\lambda$. As shown in the example reported in figure 1b, discarding information without any additional constraint can increase the amount of latent concept shift depending on the structure of the underlying data generating process. This is because discarding information is a necessary but not sufficient condition to reduce the latent concept shift.

---

[4]In contrast with the domain adaptation and generalization literature, we will consider the more general case in which $\mathbf{e}$ is represented by a vector.

## 3.2 Independence Criterion

Whenever the data selection t depends only on some observed variable $\mathbf{e}$ ($t = f(\mathbf{e}; \epsilon)$), the most intuitive approach to reduce the latent test error is to make the representation $\mathbf{z}$ independent of the environmental factors $\mathbf{e}$. This can be done by minimizing mutual information between $\mathbf{e}$ and $\mathbf{z}$: $\mathcal{R}(q_{\mathbf{z}|\mathbf{x}}) := I_{t=1}(\mathbf{e}; \mathbf{z})$. This criterion, known as *independence* or *statistical parity* in the fair classification literature (Dwork et al., 2011; Corbett-Davies et al., 2017), aims to remove any environmental information from $\mathbf{z}$, resulting in a representation that satisfies $p(\mathbf{z}|t = 1) = p(\mathbf{z}|\mathbf{e}, t = 1)$.

Despite the usefulness of this criterion in the fairness and differential privacy literature, enforcing independence does not necessarily reduce the test error (Zhao et al., 2019; Johansson et al., 2019). This is because a consistent marginal across different environments does not imply a consistent predictive distribution ($p(\mathbf{y}|\mathbf{z}, \mathbf{e}, t = 1) \neq p(\mathbf{y}|\mathbf{z}, t = 1)$), and enforcing independence may even increase the latent concept shift and the test error (as shown in figure 3).

## 3.3 Sufficiency Criterion

Instead of enforcing a property on the marginal feature distribution, one can consider stable properties of the joint distribution of features $\mathbf{z}$ and labels $\mathbf{y}$. The requirement of creating a representation that yields a stable classifier for different values of the environmental factor $\mathbf{e}$ can be captured by the following regularization: $\mathcal{R}(q_{\mathbf{z}|\mathbf{x}}) := I_{t=1}(\mathbf{y}; \mathbf{e}|\mathbf{z})$. Intuitively minimizing $I_{t=1}(\mathbf{y}; \mathbf{e}|\mathbf{z})$ corresponds to minimizing the distance between the predictive distribution $p(\mathbf{y}|\mathbf{e}, \mathbf{z}, t = 1)$ for each one of the observed environmental conditions. We can show the following:

**Proposition 4.** *Whenever the selection* t *can be expressed as a function of* $\mathbf{e}$ *and some independent noise* $\epsilon$*, the latent concept shift induced by a representation* $\mathbf{z}$ *of* $\mathbf{x}$ *is upper-bounded by* $I(\mathbf{y}; \mathbf{e}|\mathbf{z})$*:*

$$\exists f : \ t = f(\mathbf{e}, \epsilon) \implies I(\mathbf{y}; t|\mathbf{z}) \leq I(\mathbf{y}; \mathbf{e}|\mathbf{z}). \tag{11}$$

In other words, for a given selection t, it is possible to remove the effect of concept shift by enforcing the *sufficiency* constraint using the variable (or variables) $\mathbf{e}$ that are responsible for that selection.

The result from proposition 4 is applicable only if the two following conditions are met: (i) a sufficient representation must exist (Koyama & Yamaguchi, 2020); (ii) enforcing sufficiency on the selected train distribution results in sufficiency for the overall joint distribution ($I_{t=1}(\mathbf{y}; \mathbf{e}|\mathbf{z}) = 0 \implies I(\mathbf{y}; \mathbf{e}|\mathbf{z}) = 0$) (Rosenfeld et al., 2020). Even if assumptions (i) and (ii) are often acceptable in practice, lack of environment variety at training time or direct dependencies between environmental factors and targets (such as in the y-CMNIST dataset in section 4.2) can compromise the effectiveness of the sufficiency criterion. An in depth discussion with a simple example is reported in appendix G.

## 3.4 Separation Criterion

The last family of objectives and design principles includes approaches that aim to capture the stability in the latent feature distribution when the target is observed across different environmental conditions $p(\mathbf{z}|\mathbf{y}) = p(\mathbf{z}|\mathbf{y}, \mathbf{e})$ (Chouldechova, 2017; Li et al., 2018b). This requirement can be enforced by minimizing the dependency between environmental factors and the representation when the label is observed: $\mathcal{R}(q_{\mathbf{z}|\mathbf{x}}) := I_{t=1}(\mathbf{e}; \mathbf{z}|\mathbf{y})$. The resulting *separation* criterion (since $\mathbf{y}$ separates $\mathbf{z}$ and $\mathbf{e}$) can be used to identify stable properties of the joint distributions even when the selection t depends on both targets $\mathbf{y}$ and environmental factors $\mathbf{e}$:

**Proposition 5.** *If the selection* t *can be expressed as a function of* $\mathbf{e}$*,* $\mathbf{y}$ *and some independent noise* $\epsilon$*, the latent concept shift of a representation* $\mathbf{z}$ *of* $\mathbf{x}$ *is upper-bounded by the sum of prior-shift* $I(\mathbf{y}; t)$ *and* $I(\mathbf{e}; \mathbf{z}|\mathbf{y})$*:*

$$\exists f : \ t = f(\mathbf{e}, \mathbf{y}, \epsilon) \implies I(\mathbf{y}; t|\mathbf{z}) \leq I(\mathbf{y}; t) + I(\mathbf{e}; \mathbf{z}|\mathbf{y}). \tag{12}$$

When selection t and targets $\mathbf{y}$ are marginally independent, proposition 5 guarantees that the latent concept shift of a representation that enforces separation is zero. Furthermore, whenever the marginal distribution $p(\mathbf{y}|t = 0)$ is known, it is possible to adjust the prediction of the latent classifier on the test distribution[5].

---

[5]Further details on the re-weighting procedure can be found in appendix B.8

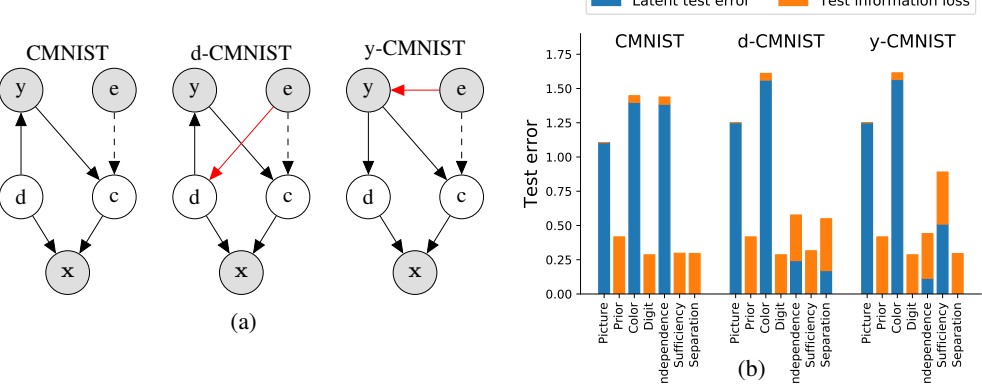

(a)

(b)

Figure 2: Graphical models (a) and error components (b) for the CMNIST, d-CMNIST and y-CMNIST data distributions. (a) Dashed lines are used to underline marginal independence between color c and environment e, while red arrows denote dependencies added to the original CMNIST distribution. (b) Models trained with strong regularization ($\lambda \approx 10^7$) for the different criteria are compared against the classifiers trained using only color, digit, picture, or prior information. The colors show the proportion of the test error (in nats) due to the predictive information loss ($I_{t=0}(\mathbf{x}; \mathbf{y}|\mathbf{z})$, in orange) and the latent test error ($D_{\mathrm{KL}}(p_{\mathbf{y}|\mathbf{z}}^{\mathrm{t}=0}||q_{\mathbf{y}|\mathbf{z}})$, in blue) according to the decomposition in equation 7.

Similarly to the other criteria, one needs to assume that enforcing separation on train ($I_{t=1}(\mathbf{e}; \mathbf{z}|\mathbf{y}) = 0$) suffices to guarantee $I(\mathbf{e}; \mathbf{z}|\mathbf{y}) = 0$. Although a representation that enforces separation always exists, the effectiveness of the separation criteria depends on the data-generating process, since the requirement could be exclusively satisfied by a constant representation.

## 4 Experiments

We evaluate the effectiveness of the criteria presented in section 3 and some of their most popular implementations on multiple versions of the CMNIST dataset (Arjovsky et al., 2019) produced by altering the data-generating process (figure 2a) to underline the shortcomings of the different methods.

The main advantage of using a synthetic dataset such as CMNIST lies in the possibility to directly optimize for the different criteria using differentiable discrete mutual information measurements. This is possible since the joint occurrence of color and digit $\hat{\mathbf{x}} := [\mathrm{c}, \mathrm{d}]$ is a low-dimensional discrete sufficient statistic of the pictures $\mathbf{x}$, and, consequently, all the mutual information quantities of interest involving $\mathbf{x} \in [0, 1]^{28 \times 28 \times 2}$ can be computed using $\hat{\mathbf{x}} \in \{0, 1\} \times \{0, \dots, 9\}$ instead. Further details regarding the direct optimization of the criteria are reported in appendix D.

In figure 3, the theoretical performance for each criterion is compared against the results obtained by training different models on the pictures $\mathbf{x}$ using neural network architectures to parametrize the encoder $q(\mathbf{z}|\mathbf{x})$ and classifier $q(\mathbf{y}|\mathbf{z})$ for different regularization strength $\lambda$. For the comparison, we consider diverse popular models designed according to the criteria defined in section 3:

- Variational Information Bottleneck (VIB) (Alemi et al., 2017): a variational tractable approximation of the **Information Bottleneck** criterion;

- Domain Adversarial Neural Network (DANN) (Ganin et al., 2016): an adversarial model based on a min-max game with discriminator $d(\mathbf{e}|\mathbf{z})$ that is optimized to predict environment information from the representation $\mathbf{z}$ to enforce the **Independence** criterion;

- Invariant Risk Minimization (IRM) (Arjovsky et al., 2019): A model designed following the **Sufficiency** criterion that aims to create a representation from which the same latent classifier is simultaneously optimal on all environments.

- Conditional Domain Adversarial Neural Network (CDANN) (Li et al., 2018b): an adversarial approximation of the **Separation** criterion in which, analogously to DANN, a discriminator tries to predict the environment when the representation $\mathbf{z}$ and the true label $\mathbf{y}$ are given.

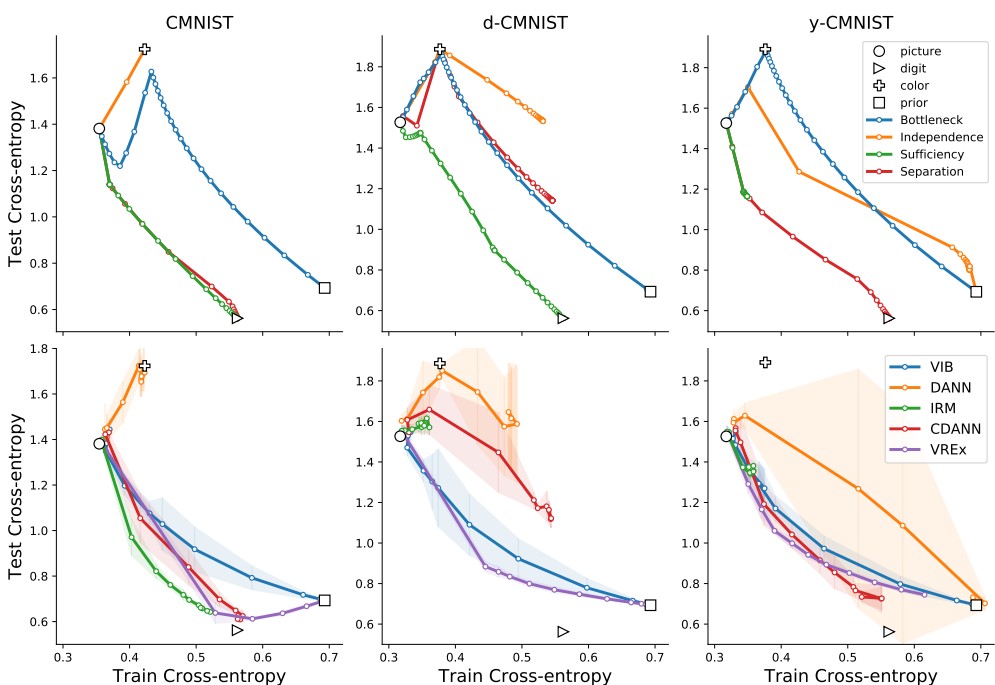

Figure 3: Comparison of training cross-entropy (x-axis) and test cross-entropy (y-axis) on the CMNIST, d-CMNIST and y-CMNIST datasets for representations trained using different criteria directly (top row) and corresponding approximations/model (bottom row). Each trajectory describes the trade-off between the two errors for a wide range of regularization strength values $\lambda$. The white symbols in each plot represent the error of models that consider only color ($p(\mathbf{y}|\mathbf{c}, \mathrm{t} = 1)$), digit ($p(\mathbf{y}|\mathbf{d}, \mathrm{t} = 1)$), picture ($p(\mathbf{y}|\mathbf{x}, \mathrm{t} = 1)$) or prior ($p(\mathbf{y}|\mathrm{t} = 1)$) information. The shaded area represents the standard deviation observed across three runs with different seeds.

- Variance-based Risk Extrapolation (VREx) (Krueger et al., 2020): a model designed following the **Sufficiency** and **Independence** criteria, based on the minimization of the training error variance across the different environments;

For a better comparison, all the models use the same encoder and classifier neural network architectures. Each model has been trained by slowly increasing the regularization strength after an initial pre-training with small $\lambda$. Further details regarding the neural network architectures, objectives, optimization and specific hyper-parameters can be found in appendix $E$.[6]

## 4.1 Evaluation metric

The measure of test error, information loss and latent test error reported in figure 2b can be computed only for discrete variables. For neural network models the training and test error defined in section 2 can be estimated up to a constant entropy by considering the expected negative log-likelihood (empirical cross-entropy):

$$D_{\mathrm{KL}}(p_{\mathbf{y}|\mathbf{x}}^{\mathrm{t}=1}||q_{\mathbf{y}|\mathbf{x}}) \approx -\frac{1}{N}\sum_{i=1}^{N}\log q(\mathbf{y} = \boldsymbol{y}_i|\mathbf{x} = \boldsymbol{x}_i) - \underbrace{H_{\mathrm{t}=1}(\mathbf{y}|\mathbf{x})}_{\text{constant}} \quad (13)$$

The samples $\boldsymbol{x}_i, \boldsymbol{y}_i$ are obtained from $p(\mathbf{x}, \mathbf{y}|\mathrm{t} = 1)$ and $p(\mathbf{x}, \mathbf{y}|\mathrm{t} = 0)$ for train and test cross-entropy respectively. By computing the training and test cross-entropy values for different regularization strength $\lambda$, each model defines a trajectory from a maximum likelihood solution ($\lambda = 0$) to the results obtained when the corresponding independence constraint is enforced (large $\lambda$), as shown in figure 3. Contrarily to accuracy measurements, the use of cross-entropy allows us to detect under-confident or over-confident predictive distributions.

---

[6]The implementation of the models in this work is available at `https://github.com/mfederici/dsit`

## 4.2 Datasets

The two variants of the CMNIST dataset considered in this work have been designed by minimally changing the original distribution to underline the strengths and weaknesses of the different criteria. Across different experiments, each MNIST picture $\mathbf{x}$ of a digit d is associated with a color c that depends on a binary target y and an environment e. The d-CMNIST and y-CMNIST datasets are created by adding a dependency from the environment to digit d (d-CMNIST) and label y (y-CMNIST), which results in a correlation between targets and environment ($I(\text{e}; \text{y}) > 0$). Further details regarding the conditional distributions used to produce the different datasets can be found in appendix C.

The strong correlation between color and label across the different CMNIST versions can be seen as an artifact introduced by the selection $\text{t} = f(\text{e})$, and models that consider only digit information (white triangles in figure 3) outperform the ones that capture color information (white crosses) or both (white circle) in terms of test cross-entropy (y-axis).

**CMNIST** The CMNIST dataset was originally designed to underline the weaknesses of maximum-likelihood and Empirical Risk Minimization strategies (Arjovsky et al., 2019). The results displayed in the first column of figure 3 confirm that both the sufficiency and separation criteria manage to effectively reduce the test cross entropy error for sufficiently strong regularization $\lambda$, minimizing the latent test error while retaining more predictive information than a constant representation (first column in figure 2b). On the CMNIST dataset, most of the models in analysis closely follow the trajectory estimated by optimizing the corresponding criterion directly. The trajectory defined by the VIB model differs from the one described by the Information Bottleneck criterion since, in absence of additional constraints, the nature of the information that is discarded (either color, digit or style) depends on the inductive bias introduced by the specific architecture.

**d-CMNIST** The d-CMNIST dataset adds a dependency between environment e and digit d, increasing the frequency of specific digits for some environments. Digit information makes environment and label conditionally independent (y and e are d-separated in figure 2a). As claimed in Arjovsky et al. (2019), models based on the sufficiency criterion manage to reduce the latent test error (second column of figure 2b), while the separation criterion fails because of the direct dependency between environment and digits. The independence criterion does not improve the test performance since both color and digit correlate with the environment. Both DANN and CDANN architectures results in trajectories that are similar to the ones obtained optimizing for the corresponding criteria. Note that enforcing separation or independence does not improve upon the result that can be obtained by blindly discarding information using VIB. Despite the effectiveness of the sufficiency criterion, the model trained with the IRM objective lies far from the optimal solution. We believe that this is due to the relaxations and approximation introduced by the optimized objective as discussed in appendix F.

**y-CMNIST** Adding a dependency between environment e and label y simulates the scenario in which some labels are more frequent in some environments. The arrow between d and y flips when compared to the d-CMNIST dataset to represent stable $p(\text{d}|\text{y})$ across different environments. The corresponding y-CMNIST dataset includes a path from e to y that can not be blocked since no representation $\mathbf{z}$ can achieve sufficiency ($I(\text{e}; \text{y}|\mathbf{z}) = 0$). Despite the optimality of the separation criterion (red line, third column of figure 3), the model trained with the CDANN objective struggles to minimize the test error due to instabilities of the adversarial training procedure, as discussed in appendix F. Once again, the trajectory described by the VREx objective is more favorable when compared to VIB, while the model trained using the IRM objective is far from optimality. Figure 2b confirms that, on the y-CMNIST dataset, the separation criterion is the only one that minimizes the latent test error without discarding the entirety of the picture information. This underlines that the effectiveness of sufficiency and separation criteria strongly depends on the structure of the underlying graphical model.

## 5 Related work

The problem of distribution (or dataset) shift (Quionero-Candela et al., 2009; Koh et al., 2020) has been explored in different areas in the machine learning literature ranging domain adaptation and generalization (Wang & Deng, 2018) to sub-population shift and fair classification (Mehrabi et al.,

2019; Santurkar et al., 2020). Although goals, availability of test features and environments at training time, and data selection criteria (Koh et al., 2020) may differ, most recent approaches focus on extracting features with some desired properties through three different families of objectives.

The independence criterion has been used to find the transformation which minimizes the distance between the distribution of the encoded features across different environments by using adversarial training (Ganin et al., 2016; Xie et al., 2017; Li et al., 2018a), features adjustment (Lum & Johndrow, 2016), kernel methods (Muandet et al., 2013), or variational approaches (Louizos et al., 2016; Moyer et al., 2018; Ilse et al., 2020). Despite the success of the independence criterion, Zhao et al. (2019); Johansson et al. (2019) have shown that enforcing stability of the marginal feature distribution is not sufficient to guarantee generalization.

Li et al. (2018b) extends the independence criterion using adversarial training to create representations that are independent of the environment when the label is observed. The corresponding separation criterion has been explored in the context of fair classification as *equalized odds* (Hardt et al., 2016) and some tractable relaxations (Darlington, 1971; Woodworth et al., 2017; Chouldechova, 2017).

The idea of considering features that lead to a consistent predictive distribution has been explored in the causality literature (Peters et al., 2016; Magliacane et al., 2018) for linear models as a feature selection criteria. Other work approached the problem in the non-linear case by considering gradients of the classifier (Arjovsky et al., 2019; Koyama & Yamaguchi, 2020; Parascandolo et al., 2020) or penalizing the variance of error (Krueger et al., 2020; Xie et al., 2020) across different environments. Other relaxations of the sufficiency criterion such as as *predictive parity* (Chouldechova, 2017) have been considered in the fairness literature.

Despite the promising directions of research, Gulrajani & Lopez-Paz (2020) have shown that the performance of the most popular approaches in the literature strongly depends on the hyper-parameter tuning strategy, underlying the problem of the lack of (i) standardized benchmark procedures and (ii) clarity regarding guarantees and hidden assumptions for the different algorithms. Although the problem of distribution shift has been widely studied in the classic domain generalization literature (Mansour et al., 2009; Schölkopf et al., 2012; Gong et al., 2016), this work aims to characterize the problem from a representation learning and information theoretical perspective, providing different insights and identifying potential issues and benefits of several popular models.

## 6 Discussion and Conclusion

In this work, we characterize an information-theoretic framework to analyze the distribution shift problem by relating it to the train and out of distribution test error. We identify two sources of errors for models based on latent representations, and we show that different criteria explored in literature can be seen as different strategies to minimize concept shift in the latent space using extra information about the data selection procedure. We demonstrate both theoretically and empirically that their effectiveness depends on the structure of the underlying graphical model with respect to the observed variables, training data, and data selection criteria other than the chosen approximation and relaxations for optimization.

Although test information loss and latent test error are challenging to estimate for real-world datasets, we argue that the presented analysis can be useful to better understand and mitigate the effect of selection bias and distribution shift in machine learning models. An accurate estimation of the error decomposition reported in this work could be further used to guide processes of causal discovery.

## Acknowledgments and Disclosure of Funding

We thank Sindy Löwe and David Ruhe for their insightful comments and feedback. This work was supported by the Microsoft Research PhD Scholarship Programme.

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
