# A Notation

For a given joint distribution $p(\mathbf{x}, \mathbf{y}, \mathbf{e})$ on $\mathbf{x}$, $\mathbf{y}$ and $\mathbf{e}$, a binary selection $\mathbf{t} = f(\mathbf{x}, \mathbf{y}, \mathbf{e}, \epsilon)$, and a conditional distribution $q(\mathbf{y}|\mathbf{x})$ we use the following notation

## A.1 Conditional Kullback-Leibler divergence

$$D_{\mathrm{KL}}(p(\mathbf{y}|\mathbf{x})||q(\mathbf{y}|\mathbf{x})) := \mathbb{E}_{\boldsymbol{x},\boldsymbol{y}\sim p(\mathbf{x},\mathbf{y})}\left[\log\frac{p(\mathbf{y}=\boldsymbol{y}|\mathbf{x}=\boldsymbol{x})}{q(\mathbf{y}=\boldsymbol{y}|\mathbf{x}=\boldsymbol{x})}\right]. \tag{14}$$

Note that the expectation is always considered with respect to the joint distribution for the first argument of the KL-divergence. The notation $D_{\mathrm{KL}}(p_{\mathbf{y}|\mathbf{x}}||q_{\mathbf{y}|\mathbf{x}})$ will be used to abbreviate the same quantity.

## A.2 Jensen-Shannon divergence

$$D_{\mathrm{JSD}}(p(\mathbf{x})||q(\mathbf{x})) := \frac{1}{2}D_{\mathrm{KL}}(p(\mathbf{x})||m(\mathbf{x})) + \frac{1}{2}D_{\mathrm{KL}}(q(\mathbf{x})||m(\mathbf{x})), \tag{15}$$

with $m(\mathbf{x}) = 1/2\, p(\mathbf{x}) + 1/2\, q(\mathbf{x})$.

## A.3 Mutual Information

$$I(\mathbf{x}; \mathbf{y}) := D_{\mathrm{KL}}(p(\mathbf{x}, \mathbf{y})||p(\mathbf{x})p(\mathbf{y})) \tag{16}$$
$$= D_{\mathrm{KL}}(p(\mathbf{y}|\mathbf{x})||p(\mathbf{y})) \tag{17}$$
$$= D_{\mathrm{KL}}(p(\mathbf{x}|\mathbf{y})||p(\mathbf{x})). \tag{18}$$

The subscript $\mathbf{t} = 1$ is used to indicate that both joint and marginal distribution are conditioned on $\mathbf{t} = 1$:

$$I_{\mathbf{t}=1}(\mathbf{x}; \mathbf{y}) := D_{\mathrm{KL}}(p(\mathbf{x}, \mathbf{y}|\mathbf{t}=1)||p(\mathbf{x}|\mathbf{t}=1)p(\mathbf{y}|\mathbf{t}=1)) \tag{19}$$

Conditional mutual information is defined as:

$$I(\mathbf{x}; \mathbf{y}|\mathbf{e}) := D_{\mathrm{KL}}(p(\mathbf{x}, \mathbf{y}|\mathbf{e})||p(\mathbf{x}|\mathbf{e})p(\mathbf{y}|\mathbf{e})) \tag{20}$$
$$:= D_{\mathrm{KL}}(p(\mathbf{y}|\mathbf{x}, \mathbf{e})||p(\mathbf{y}|\mathbf{e})) \tag{21}$$
$$:= D_{\mathrm{KL}}(p(\mathbf{x}|\mathbf{y}, \mathbf{e})||p(\mathbf{x}|\mathbf{e})). \tag{22}$$

Note that all the mutual information terms in this work are implicitly expressed in terms of the distribution $p$.

# B Proofs and Additional details

The proofs for the statement in the main text are reported in this section.

## B.1 Proof for Proposition 1

*Proof.* Consider $D_{\mathrm{KL}}(p(\mathbf{y}|\mathbf{x})||q(\mathbf{y}|\mathbf{x}))$. By multiplying and dividing by $p(\mathbf{y}|\mathbf{x}, \mathbf{t})$, we obtain:

$$D_{\mathrm{KL}}(p(\mathbf{y}|\mathbf{x})||q(\mathbf{y}|\mathbf{x})) = \mathbb{E}_{\boldsymbol{x},\boldsymbol{y}\sim p(\mathbf{x},\mathbf{y})}\left[\log\frac{p(\mathbf{y}=\boldsymbol{y}|\mathbf{x}=\boldsymbol{x})}{q(\mathbf{y}=\boldsymbol{y}|\mathbf{x}=\boldsymbol{x})}\right]$$
$$= \mathbb{E}_{\boldsymbol{x},\boldsymbol{y},t\sim p(\mathbf{x},\mathbf{y},\mathbf{t})}\left[\log\frac{p(\mathbf{y}=\boldsymbol{y}|\mathbf{x}=\boldsymbol{x},\mathbf{t}=t)}{q(\mathbf{y}=\boldsymbol{y}|\mathbf{x}=\boldsymbol{x})}\frac{p(\mathbf{y}=\boldsymbol{y}|\mathbf{x}=\boldsymbol{x})}{p(\mathbf{y}=\boldsymbol{y}|\mathbf{x}=\boldsymbol{x},\mathbf{t}=t)}\right]$$
$$= D_{\mathrm{KL}}(p(\mathbf{y}|\mathbf{x},\mathbf{t})||q(\mathbf{y}|\mathbf{x})) - I(\mathbf{y};\mathbf{t}|\mathbf{x}). \tag{23}$$

Splitting the expectation on t, we express the first term as:

$$D_{\mathrm{KL}}(p(\mathbf{y}|\mathbf{x},\mathrm{t})||q(\mathbf{y}|\mathbf{x})) = p(\mathrm{t}=1)\mathbb{E}_{\boldsymbol{x},\boldsymbol{y}\sim p(\mathbf{x},\mathbf{y}|\mathrm{t}=1)}\left[\log\frac{p(\mathbf{y}=\boldsymbol{y}|\mathbf{x}=\boldsymbol{x},\mathrm{t}=1)}{q(\mathbf{y}=\boldsymbol{y}|\mathbf{x}=\boldsymbol{x})}\right]$$

$$+ p(\mathrm{t}=0)\,\mathbb{E}_{\boldsymbol{x},\boldsymbol{y}\sim p(\mathbf{x},\mathbf{y}|\mathrm{t}=0)}\left[\log\frac{p(\mathbf{y}=\boldsymbol{y}|\mathbf{x}=\boldsymbol{x},\mathrm{t}=0)}{q(\mathbf{y}=\boldsymbol{y}|\mathbf{x}=\boldsymbol{x})}\right]$$

$$= p(\mathrm{t}=1)D_{\mathrm{KL}}(p(\mathbf{y}|\mathbf{x},\mathrm{t}=1)||q(\mathbf{y}|\mathbf{x})) + p(\mathrm{t}=0)D_{\mathrm{KL}}(p(\mathbf{y}|\mathbf{x},\mathrm{t}=0)||q(\mathbf{y}|\mathbf{x}))$$

$$= p(\mathrm{t}=1)D_{\mathrm{KL}}(p_{\mathbf{y}|\mathbf{x}}^{\mathrm{t}=1}||q_{\mathbf{y}|\mathbf{x}}) + p(\mathrm{t}=0)D_{\mathrm{KL}}(p_{\mathbf{y}|\mathbf{x}}^{\mathrm{t}=0}||q_{\mathbf{y}|\mathbf{x}}). \tag{24}$$

Since the KL-divergence between two distributions is always positive, using the result from equations 23 and 24, we have:

$$I(\mathbf{y};\mathrm{t}|\mathbf{x}) \leq D_{\mathrm{KL}}(p(\mathbf{y}|\mathbf{x},\mathrm{t})||q(\mathbf{y}|\mathbf{x}))$$

$$= p(\mathrm{t}=1)D_{\mathrm{KL}}(p_{\mathbf{y}|\mathbf{x}}^{\mathrm{t}=1}||q_{\mathbf{y}|\mathbf{x}}) + p(\mathrm{t}=0)D_{\mathrm{KL}}(p_{\mathbf{y}|\mathbf{x}}^{\mathrm{t}=0}||q_{\mathbf{y}|\mathbf{x}})$$

$$\leq (1-\alpha)\left(D_{\mathrm{KL}}(p_{\mathbf{y}|\mathbf{x}}^{\mathrm{t}=1}||q_{\mathbf{y}|\mathbf{x}}) + D_{\mathrm{KL}}(p_{\mathbf{y}|\mathbf{x}}^{\mathrm{t}=0}||q_{\mathbf{y}|\mathbf{x}})\right), \tag{25}$$

with $\alpha := \min\{p(\mathrm{t}=0),p(\mathrm{t}=1)\}$.

Whenever $\alpha = 1$, the inequality in equation 25 holds since $I(\mathbf{y};\mathrm{t}|\mathbf{x}) = 0$ and KL-divergence is always positive. For $\alpha < 1$ we can divide both sides by $1-\alpha$, obtaining:

$$D_{\mathrm{KL}}(p_{\mathbf{y}|\mathbf{x}}^{\mathrm{t}=1}||q_{\mathbf{y}|\mathbf{x}}) + D_{\mathrm{KL}}(p_{\mathbf{y}|\mathbf{x}}^{\mathrm{t}=0}||q_{\mathbf{y}|\mathbf{x}}) \geq \frac{1}{1-\alpha}I(\mathbf{y};\mathrm{t}|\mathbf{x})$$

$\square$

## B.2    Proof for Proposition 2

*Proof.* Using Jensen's inequality we have:

$$D_{\mathrm{KL}}(p_{\mathbf{y}|\mathbf{x}}^{\mathrm{t}=1}||q_{\mathbf{y}|\mathbf{x}}) = \mathbb{E}_{\boldsymbol{x},\boldsymbol{y}\sim(\mathbf{x},\mathbf{y}|\mathrm{t}=1)}\left[\log\frac{p(\mathbf{y}=\boldsymbol{y}|\mathbf{x}=\boldsymbol{x},\mathrm{t}=1}{\mathbb{E}_{\boldsymbol{z}\sim q(\mathbf{z}|\mathbf{x}=\boldsymbol{x})}[q(\mathbf{y}=\boldsymbol{y}|\mathbf{z}=\boldsymbol{z})]}\right]$$

$$\leq^* \mathbb{E}_{\boldsymbol{x},\boldsymbol{y}\sim(\mathbf{x},\mathbf{y}|\mathrm{t}=1)}\mathbb{E}_{\boldsymbol{z}\sim q(\mathbf{z}|\mathbf{x}=\boldsymbol{x})}\left[\log\frac{p(\mathbf{y}=\boldsymbol{y}|\mathbf{x}=\boldsymbol{x},\mathrm{t}=1)}{q(\mathbf{y}=\boldsymbol{y}|\mathbf{z}=\boldsymbol{z})}\right]$$

$$= D_{\mathrm{KL}}(p_{\mathbf{y}|\mathbf{x}}^{\mathrm{t}=1}||q_{\mathbf{y}|\mathbf{z}}), \tag{26}$$

in which $^*$ holds with equality when $q(\mathbf{z}|\mathbf{x})$ is a delta distribution (deterministic encoder).

Secondly, by multiplying and dividing by $p(\mathbf{y}=\boldsymbol{y}|\mathbf{z}=\boldsymbol{z},\mathrm{t}=1)$:

$$D_{\mathrm{KL}}(p_{\mathbf{y}|\mathbf{x}}^{\mathrm{t}=1}||q_{\mathbf{y}|\mathbf{z}}) = \mathbb{E}_{\boldsymbol{x},\boldsymbol{y}\sim(\mathbf{x},\mathbf{y}|\mathrm{t}=1)}\mathbb{E}_{\boldsymbol{z}\sim q(\mathbf{z}|\mathbf{x}=\boldsymbol{x})}\left[\log\frac{p(\mathbf{y}=\boldsymbol{y}|\mathbf{x}=\boldsymbol{x},\mathrm{t}=1)}{p(\mathbf{y}=\boldsymbol{y}|\mathbf{z}=\boldsymbol{z},\mathrm{t}=1)}\frac{p(\mathbf{y}=\boldsymbol{y}|\mathbf{z}=\boldsymbol{z},\mathrm{t}=1)}{q(\mathbf{y}=\boldsymbol{y}|\mathbf{z}=\boldsymbol{z})}\right]$$

$$= I(\mathbf{y};\mathbf{x}|\mathbf{z}) + D_{\mathrm{KL}}(p_{\mathbf{y}|\mathbf{z}}^{\mathrm{t}=1}||q_{\mathbf{y}|\mathbf{z}}). \tag{27}$$

The bound for $D_{\mathrm{KL}}(p_{\mathbf{y}|\mathbf{x}}^{\mathrm{t}=0}||q(\mathbf{y}|\mathbf{x}))$ is obtained analogously. $\square$

## B.3 Proof for Proposition 3

*Proof.* We first define the average latent predictive distribution $\overline{p}(\mathbf{y}|\mathbf{z}) := \frac{1}{2}p(\mathbf{y}|\mathbf{z},\mathrm{t}=0) + \frac{1}{2}p(\mathbf{y}|\mathbf{z},\mathrm{t}=1)$, then

$$
\begin{aligned}
I(\mathbf{y};\mathrm{t}|\mathbf{z}) &= D_{\mathrm{KL}}(p(\mathbf{y}|\mathbf{z},\mathrm{t})||p(\mathbf{y}|\mathbf{z})) \\
&= p(\mathrm{t}=0)D_{\mathrm{KL}}(p_{\mathbf{y}|\mathbf{z}}^{\mathrm{t}=0}||p_{\mathbf{y}|\mathbf{z}}) + p(\mathrm{t}=1)D_{\mathrm{KL}}(p_{\mathbf{y}|\mathbf{z}}^{\mathrm{t}=1}||p_{\mathbf{y}|\mathbf{z}}) \\
&\geq 2\alpha \left( \frac{1}{2}D_{\mathrm{KL}}(p_{\mathbf{y}|\mathbf{z}}^{\mathrm{t}=0}||p_{\mathbf{y}|\mathbf{z}}) + \frac{1}{2}D_{\mathrm{KL}}(p_{\mathbf{y}|\mathbf{z}}^{\mathrm{t}=1}||p_{\mathbf{y}|\mathbf{z}}) \right) \\
&= 2\alpha\,\mathbb{E}_{t\sim Ber(1/2)}\mathbb{E}_{\mathbf{z},\mathbf{y}\sim p(\mathbf{z},\mathbf{y}|\mathrm{t}=t)}\left[ \log \frac{p(\mathbf{y}=\mathbf{y}|\mathbf{z}=\mathbf{z},\mathrm{t}=t)}{p(\mathbf{y}=\mathbf{y}|\mathbf{z}=\mathbf{z})} \right] \\
&= 2\alpha\,\mathbb{E}_{t\sim Ber(1/2)}\mathbb{E}_{\mathbf{z},\mathbf{y}\sim p(\mathbf{z},\mathbf{y}|\mathrm{t}=t)}\left[ \log \frac{p(\mathbf{y}=\mathbf{y}|\mathbf{z}=\mathbf{z},\mathrm{t}=t)}{\overline{p}(\mathbf{y}=\mathbf{y}|\mathbf{z}=\mathbf{z})} \frac{\overline{p}(\mathbf{y}=\mathbf{y}|\mathbf{z}=\mathbf{z})}{p(\mathbf{y}=\mathbf{y}|\mathbf{z}=\mathbf{z})} \right] \\
&= 2\alpha\,D_{\mathrm{JSD}}(p_{\mathbf{y}|\mathbf{z}}^{\mathrm{t}=0}||p_{\mathbf{y}|\mathbf{z}}^{\mathrm{t}=1}) + D_{\mathrm{KL}}(\overline{p}(\mathbf{y}|\mathbf{z})||p(\mathbf{y}|\mathbf{z})) \\
&\geq 2\alpha\,D_{\mathrm{JSD}}(p_{\mathbf{y}|\mathbf{z}}^{\mathrm{t}=0}||p_{\mathbf{y}|\mathbf{z}}^{\mathrm{t}=1}),
\end{aligned} \tag{28}
$$

where $Ber(1/2)$ refers to a Bernoulli distribution with parameter $p=1/2$.

Secondly, since the square root of the Jensen-Shannon divergence is a metric (Endres & Schindelin, 2003), using triangle inequality we have:

$$
\sqrt{D_{\mathrm{JSD}}(p_{\mathbf{y}|\mathbf{z}=\mathbf{z}}^{\mathrm{t}=1}||q_{\mathbf{y}|\mathbf{z}=\mathbf{z}})} + \sqrt{D_{\mathrm{JSD}}(p_{\mathbf{y}|\mathbf{z}=\mathbf{z}}^{\mathrm{t}=0}||p_{\mathbf{y}|\mathbf{z}=\mathbf{z}}^{\mathrm{t}=1})} \geq \sqrt{D_{\mathrm{JSD}}(p_{\mathbf{y}|\mathbf{z}=\mathbf{z}}^{\mathrm{t}=0}||q_{\mathbf{y}|\mathbf{z}=\mathbf{z}})}. \tag{29}
$$

Note that the conditional distributions $p_{\mathbf{y}|\mathbf{z}=\mathbf{z}}^{\mathrm{t}=0}$ and $p_{\mathbf{y}|\mathbf{z}=\mathbf{z}}^{\mathrm{t}=1}$ are defined only for latent vectors $\mathbf{z}$ that have positive probability according to $p(\mathbf{z}|\mathrm{t}=0)$ and $p(\mathbf{z}|\mathrm{t}=1)$ respectively. Therefore inequality is defined only for $\mathbf{z}$ that have simultaneously strictly positive probability on both train and test distributions.

Lastly, The Jensen-Shannon divergence can be upper-bounded by a function of the corresponding Kullback-Leibler divergence:

$$
D_{\mathrm{JSD}}(p_{\mathbf{y}|\mathbf{z}}^{\mathrm{t}=1}||q_{\mathbf{y}|\mathbf{z}}) = \frac{1}{2}D_{\mathrm{KL}}(p_{\mathbf{y}|\mathbf{z}}^{\mathrm{t}=1}||q_{\mathbf{y}|\mathbf{z}}) - D_{\mathrm{KL}}(m_{\mathbf{y}|\mathbf{z}}||q_{\mathbf{y}|\mathbf{z}}) \leq \frac{1}{2}D_{\mathrm{KL}}(p_{\mathbf{y}|\mathbf{z}}^{\mathrm{t}=1}||q_{\mathbf{y}|\mathbf{z}}), \tag{30}
$$

with $m_{\mathbf{y}|\mathbf{z}} := 1/2p(\mathbf{y}|\mathbf{z},\mathrm{t}=1) + 1/2q(\mathbf{y}|\mathbf{z})$.

Using the results from 28, 29 and 30 we have

$$
\begin{aligned}
D_{\mathrm{JSD}}(p_{\mathbf{y}|\mathbf{z}=\mathbf{z}}^{\mathrm{t}=0}||q_{\mathbf{y}|\mathbf{z}=\mathbf{z}}) &\overset{(29)}{\leq} \left( \sqrt{D_{\mathrm{JSD}}(p_{\mathbf{y}|\mathbf{z}=\mathbf{z}}^{\mathrm{t}=1}||q_{\mathbf{y}|\mathbf{z}=\mathbf{z}})} + \sqrt{D_{\mathrm{JSD}}(p_{\mathbf{y}|\mathbf{z}=\mathbf{z}}^{\mathrm{t}=0}||p_{\mathbf{y}|\mathbf{z}=\mathbf{z}}^{\mathrm{t}=1})} \right)^2 \\
&\overset{(28)}{\leq} \left( \sqrt{\frac{1}{2\alpha}I(\mathbf{y};\mathrm{t}|\mathbf{z}=\mathbf{z})} + \sqrt{D_{\mathrm{JSD}}(p_{\mathbf{y}|\mathbf{z}=\mathbf{z}}^{\mathrm{t}=0}||p_{\mathbf{y}|\mathbf{z}=\mathbf{z}}^{\mathrm{t}=1})} \right)^2 \\
&\overset{(30)}{\leq} \left( \sqrt{\frac{1}{2\alpha}I(\mathbf{y};\mathrm{t}|\mathbf{z}=\mathbf{z})} + \sqrt{\frac{1}{2}D_{\mathrm{KL}}(p_{\mathbf{y}|\mathbf{z}=\mathbf{z}}^{\mathrm{t}=1}||q_{\mathbf{y}|\mathbf{z}=\mathbf{z}})} \right)^2
\end{aligned} \tag{31}
$$

$\square$

## B.4 Upper-bounding the test error

We show two inequalities, that relate the Jensen-Shannon divergence in equation 31 to the latent test error (defined as a Kullback-Leibler divergence).

### B.4.1 Ratio of $f$-divergences

Since both Jensen-Shannon and Kullback Leibler divergence are instances of $f$-divergences, we can use the result from theorem 1 in Sason & Verdú (2016) to write the following inequality:

$$D_{\mathrm{JSD}}(p^{\mathrm{t=0}}_{\mathbf{y}|\mathbf{z}=\boldsymbol{z}}||q_{\mathbf{y}|\mathbf{z}=\boldsymbol{z}}) \geq K(p^{\mathrm{t=0}}_{\mathbf{y}|\mathbf{z}=\boldsymbol{z}}, q_{\mathbf{y}|\mathbf{z}=\boldsymbol{z}})D_{\mathrm{KL}}(p^{\mathrm{t=0}}_{\mathbf{y}|\mathbf{z}=\boldsymbol{z}}||q_{\mathbf{y}|\mathbf{z}=\boldsymbol{z}}), \tag{32}$$

with

$$K(p^{\mathrm{t=0}}_{\mathbf{y}|\mathbf{z}=\boldsymbol{z}}, q_{\mathbf{y}|\mathbf{z}=\boldsymbol{z}}) = \sup_{\boldsymbol{y}} k\left(\frac{p(\mathbf{y}=\boldsymbol{y}|\mathbf{z}=\boldsymbol{z}, \mathrm{t}=0)}{q(\mathbf{y}=\boldsymbol{y}|\mathbf{z}=\boldsymbol{z})}\right),$$

in which

$$k(r) = \begin{cases} \frac{f(r)}{g(r)} & \text{for } r \in (0,1) \cup (1,+\infty) \\ 1 & \text{for } r = 1 \end{cases},$$

and

$$f(r) = r \log r$$
$$g(r) = \frac{1}{2}\left[r \log r - (r+1)\log\frac{r+1}{2}\right]$$

### B.4.2 Pinsker-type bound

Fisrt, since the Jensen-Shannon divergence is a $f$-divergence, we can use the result from theorem 3 in Gilardoni (2010) to express a lower-bound as a function of the total variation $D_{TV}$:

$$\sqrt{D_{\mathrm{JSD}}(p^{\mathrm{t=0}}_{\mathbf{y}|\mathbf{z}=\boldsymbol{z}}||q_{\mathbf{y}|\mathbf{z}=\boldsymbol{z}})} \geq \sqrt{\frac{f''(1)}{2}}\, D_{TV}(p^{\mathrm{t=0}}_{\mathbf{y}|\mathbf{z}=\boldsymbol{z}}||q_{\mathbf{y}|\mathbf{z}=\boldsymbol{z}})$$
$$= \sqrt{2}\, D_{TV}(p^{\mathrm{t=0}}_{\mathbf{y}|\mathbf{z}=\boldsymbol{z}}||q_{\mathbf{y}|\mathbf{z}=\boldsymbol{z}}), \tag{33}$$

with $f''(1)$ as the second derivative of $f(t) = \frac{1}{2}\left[(t+1)\log\frac{2}{t+1} + t\log t\right]$ evaluated in $t = 1$.

Secondly, since the Kullback-Leibler divergence is also a $f$-divergence, we use the result of theorem 1 from Binette (2019) to upper-bound the latent test error as a function of the total variation:

$$D_{\mathrm{KL}}(p^{\mathrm{t=0}}_{\mathbf{y}|\mathbf{z}=\boldsymbol{z}}||q_{\mathbf{y}|\mathbf{z}=\boldsymbol{z}}) \leq \left(\frac{f(m(\mathbf{z}))}{1-m(\mathbf{z})} + \frac{f(M(\mathbf{z}))}{M(\mathbf{z})-1}\right) D_{TV}(p^{\mathrm{t=0}}_{\mathbf{y}|\mathbf{z}=\boldsymbol{z}}||q_{\mathbf{y}|\mathbf{z}=\boldsymbol{z}})$$
$$= \left(\frac{m(\mathbf{z})\log m(\mathbf{z})}{1-m(\mathbf{z})} + \frac{M(\mathbf{z})\log M(\mathbf{z})}{M(\mathbf{z})-1}\right) D_{TV}(p^{\mathrm{t=0}}_{\mathbf{y}|\mathbf{z}=\boldsymbol{z}}||q_{\mathbf{y}|\mathbf{z}=\boldsymbol{z}}), \tag{34}$$

with $m(\mathbf{z}) := \min_{\boldsymbol{y}} \frac{p(\mathbf{y}=\boldsymbol{y}|\mathbf{z}=\boldsymbol{z},\mathrm{t}=0)}{q(\mathbf{y}=\boldsymbol{y}|\mathbf{z}=\boldsymbol{z})}$, $M(\mathbf{z}) := \max_{\boldsymbol{y}} \frac{p(\mathbf{y}=\boldsymbol{y}|\mathbf{z}=\boldsymbol{z},\mathrm{t}=0)}{q(\mathbf{y}=\boldsymbol{y}|\mathbf{z}=\boldsymbol{z})}$, and $f(t) = t\log t$.

Combining the two bounds we obtain:

$$D_{\mathrm{KL}}(p^{\mathrm{t=0}}_{\mathbf{y}|\mathbf{z}=\boldsymbol{z}}||q_{\mathbf{y}|\mathbf{z}=\boldsymbol{z}}) \leq \frac{1}{\sqrt{2}}\left(\frac{m(\mathbf{z})\log m(\mathbf{z})}{1-m(\mathbf{z})} + \frac{M(\mathbf{z})\log M(\mathbf{z})}{M(\mathbf{z})-1}\right)\sqrt{D_{\mathrm{JSD}}(p^{\mathrm{t=0}}_{\mathbf{y}|\mathbf{z}=\boldsymbol{z}}||q_{\mathbf{y}|\mathbf{z}=\boldsymbol{z}})}. \tag{35}$$

Note that the right-hand side of this last inequality is finite whenever the support of the model $q(\mathbf{y}|\mathbf{z}=\boldsymbol{z})$ contains the support of $p(\mathbf{y}|\mathbf{z}=\boldsymbol{z}, \mathrm{t}=0)$ (and therefore $M(\mathbf{z})$ is finite). When this does not happen, both left-hand and right hand side of the inequality are unbounded.

### B.5 Bound in expression 9

*Proof.* Consider the amount of predictive information lost by encoding $\mathbf{x}$ into $\mathbf{z}$:

$$I_{\mathrm{t=1}}(\mathbf{x};\mathbf{y}|\mathbf{z}) = I_{\mathrm{t=1}}(\mathbf{zx};\mathbf{y}) - I_{\mathrm{t=1}}(\mathbf{z};\mathbf{y})$$
$$= I_{\mathrm{t=1}}(\mathbf{x};\mathbf{y}) + I_{\mathrm{t=1}}(\mathbf{z};\mathbf{y}|\mathbf{x}) - I_{\mathrm{t=1}}(\mathbf{z};\mathbf{y})$$
$$= I_{\mathrm{t=1}}(\mathbf{x};\mathbf{y}) - I_{\mathrm{t=1}}(\mathbf{z};\mathbf{y}), \tag{36}$$

in which $I_{\mathrm{t=1}}(\mathbf{z};\mathbf{y}|\mathbf{x}) = 0$ since $\mathbf{z}$ depends only on $\mathbf{x}$.

Since $I(\mathbf{x}; \mathbf{y})$ is constant in $q(\mathbf{z}|\mathbf{x})$:

$$\min_{q(\mathbf{z}|\mathbf{x})} I_{t=1}(\mathbf{x}; \mathbf{y}|\mathbf{z}) = I(\mathbf{x}; \mathbf{y}) - \max_{q(\mathbf{z}|\mathbf{x})} I_{t=1}(\mathbf{y}; \mathbf{z}). \tag{37}$$

The upper-bound is derived considering $q(\mathbf{y}|\mathbf{z})$ as a variational distribution. For any $q(\mathbf{y}|\mathbf{z})$:

$$\begin{aligned}
I_{t=1}(\mathbf{x}; \mathbf{y}|\mathbf{z}) &= D_{\mathrm{KL}}(p_{\mathbf{y}|\mathbf{x}}^{t=1} \| p_{\mathbf{y}|\mathbf{z}}^{t=1}) \\
&= D_{\mathrm{KL}}(p_{\mathbf{y}|\mathbf{x}}^{t=1} \| q_{\mathbf{y}|\mathbf{z}}) - D_{\mathrm{KL}}(p_{\mathbf{y}|\mathbf{z}}^{t=1} \| q_{\mathbf{y}|\mathbf{z}}) \\
&\leq D_{\mathrm{KL}}(p_{\mathbf{y}|\mathbf{x}}^{t=1} \| q_{\mathbf{y}|\mathbf{z}}),
\end{aligned} \tag{38}$$

and in particular

$$I_{t=1}(\mathbf{x}; \mathbf{y}|\mathbf{z}) \leq \min_{q(\mathbf{y}|\mathbf{z})} D_{\mathrm{KL}}(p_{\mathbf{y}|\mathbf{x}}^{t=1} \| q_{\mathbf{y}|\mathbf{z}}), \tag{39}$$

in which equality holds for $q(\mathbf{y}|\mathbf{z}) = p(\mathbf{y}|\mathbf{z}, t = 1)$. $\qquad\square$

## B.6    Proof for Proposition 4

*Proof.* Consider the latent concept shift $I(\mathbf{y}; t|\mathbf{z})$. Using the chain rule of mutual information, we obtain:

$$\begin{aligned}
I(\mathbf{y}; t|\mathbf{z}) &= I(\mathbf{y}; et|\mathbf{z}) - I(\mathbf{y}; e|\mathbf{z}t) \\
&\leq I(\mathbf{y}; et|\mathbf{z}) \\
&= I(\mathbf{y}; e|\mathbf{z}) + I(\mathbf{y}; t|\mathbf{z}e).
\end{aligned} \tag{40}$$

Whenever t can be expressed as a function of $\mathbf{e}$ and some independent noise $\epsilon$, we have:

$$\begin{aligned}
t &= f(\mathbf{e}, \epsilon) \\
&\implies p(t|\mathbf{e}) = p(t|\mathbf{e}, \mathbf{z}) = p(t|\mathbf{e}, \mathbf{z}, \mathbf{y}) \\
&\implies D_{\mathrm{KL}}(p(t|\mathbf{e}, \mathbf{z}) \| p(t|\mathbf{e}, \mathbf{z}, \mathbf{y})) = 0 \\
&\iff I(\mathbf{y}; t|\mathbf{z}e) = 0.
\end{aligned} \tag{41}$$

Therefore, we conclude:

$$I(\mathbf{y}; t|\mathbf{z}) \leq I(\mathbf{y}; e|\mathbf{z}) \tag{42}$$

$\qquad\square$

## B.7    Proof for Proposition 5

*Proof.* Consider the latent concept shift $I(\mathbf{y}; t|\mathbf{z})$. Using the chain rule of mutual information, we obtain:

$$\begin{aligned}
I(\mathbf{y}; t|\mathbf{z}) &= I(\mathbf{y}\mathbf{z}; t) - I(t; \mathbf{z}) \\
&\leq I(\mathbf{y}\mathbf{z}; t) \\
&= I(\mathbf{y}; t) + I(t; \mathbf{z}|\mathbf{y})
\end{aligned} \tag{43}$$

Then, we express $I(t; \mathbf{z}|\mathbf{y})$ as a function of $I(e; \mathbf{z}|\mathbf{y})$:

$$\begin{aligned}
I(t; \mathbf{z}|\mathbf{y}) &= I(et; \mathbf{z}|\mathbf{y}) - I(e; \mathbf{z}|\mathbf{y}t) \\
&\leq I(et; \mathbf{z}|\mathbf{y}) \\
&= I(e; \mathbf{z}|\mathbf{y}) + I(t; \mathbf{z}|\mathbf{y}e).
\end{aligned} \tag{44}$$

Whenever t can be expressed as a function of $\mathbf{e}$, $\mathbf{y}$ and some independent noise $\epsilon$, we have:

$$\begin{aligned}
\exists f &: t = f(\mathbf{e}, \mathbf{y}, \epsilon) \\
&\implies p(t|\mathbf{e}, \mathbf{y}) = p(t|\mathbf{e}, \mathbf{y}, \mathbf{z}) \\
&\implies D_{\mathrm{KL}}(p(t|\mathbf{e}, \mathbf{y}, \mathbf{z}) \| p(t|\mathbf{e}, \mathbf{y})) = 0 \\
&\iff I(t; \mathbf{z}|\mathbf{y}e) = 0.
\end{aligned} \tag{45}$$

Using the results from equations 43, 44 and 45:

$$\begin{aligned}
I(\mathbf{y}; t|\mathbf{z}) &\leq I(\mathbf{y}; t) + I(t; \mathbf{z}|\mathbf{y}) \\
&\leq I(\mathbf{y}; t) + I(e; \mathbf{z}|\mathbf{y}).
\end{aligned} \tag{46}$$

$\qquad\square$

## B.8 Correcting for prior shift

When the separation constraint $I(\mathbf{e}; \mathbf{z}|\mathbf{y}) = 0$ is enforced the *reverse concept shift* $I(\mathrm{t}; \mathbf{z}|\mathbf{y})$, which represents how much the distribution $p(\mathbf{z}|\mathbf{y})$ changes as a result of the selection, is zero (equations 44 and 45). Considering the stability of $p(\mathbf{z}|\mathbf{y})$, we can express the test predictive distribution $p(\mathbf{y}|\mathbf{z}, \mathrm{t} = 0)$ as a function of the train one using Bayes rule:

$$
\begin{aligned}
p(\mathbf{y}|\mathbf{z}, \mathrm{t} = 0) &= \frac{p(\mathbf{z}|\mathbf{y}, \mathrm{t} = 0)p(\mathbf{y}|\mathrm{t} = 0)}{p(\mathbf{z}|\mathrm{t} = 0)} \\
&=^* \frac{p(\mathbf{z}|\mathbf{y}, \mathrm{t} = 1)p(\mathbf{y}|\mathrm{t} = 0)}{p(\mathbf{z}|\mathrm{t} = 0)} \\
&= \frac{\frac{p(\mathbf{y}|\mathbf{z}, \mathrm{t} = 1)p(\mathbf{z}|\mathrm{t} = 1)}{p(\mathbf{y}|\mathrm{t} = 1)}p(\mathbf{y}|\mathrm{t} = 0)}{p(\mathbf{z}|\mathrm{t} = 0)} \\
&= p(\mathbf{y}|\mathbf{z}, \mathrm{t} = 1) \underbrace{\frac{p(\mathbf{y}|\mathrm{t} = 0)}{p(\mathbf{y}|\mathrm{t} = 1)}}_{r(\mathbf{y})} \underbrace{\frac{p(\mathbf{z}|\mathrm{t} = 1)}{p(\mathbf{z}|\mathrm{t} = 0)}}_{1/Z},
\end{aligned} \tag{47}
$$

where $^*$ uses $p(\mathbf{z}|\mathbf{y}, \mathrm{t} = 0) = p(\mathbf{z}|\mathbf{y}, \mathrm{t} = 1)$. The ratio $r(\mathbf{y})$ represents the fraction of the marginal probabilities of $\mathbf{y}$, while $Z$ is a normalization constant.

We define the corrected model $q'(\mathbf{y}|\mathbf{z})$ following the result from equation 47:

$$
\hat{q}'(\mathbf{y}|\mathbf{z}) := \frac{1}{Z}q(\mathbf{y}|\mathbf{z})r(\mathbf{y}). \tag{48}
$$

The test error for the model $q'(\mathbf{y}|\mathbf{z})$ for a representation $z$ that has positive support in both train and test ($p(\mathbf{z}|\mathrm{t} = 1) > 0$, $p(\mathbf{z}|\mathrm{t} = 0) > 0$) is determined by the train latent test error:

$$
\begin{aligned}
D_{\mathrm{KL}}(p_{\mathbf{y}|\mathbf{z}=z}^{\mathrm{t}=0}||q'_{\mathbf{y}|\mathbf{z}=z}) &= \mathbb{E}_{\boldsymbol{y}\sim p(\mathbf{y}|\mathbf{z}=z, \mathrm{t}=0)}\left[\log \frac{p(\mathbf{y} = \boldsymbol{y}|\mathbf{z} = z, \mathrm{t} = 0)}{q'(\mathbf{y} = \boldsymbol{y}|\mathbf{z} = z)}\right] \\
&= \mathbb{E}_{\boldsymbol{y}\sim p(\mathbf{y}|\mathbf{z}=z, \mathrm{t}=0)}\left[\log \frac{p(\mathbf{y} = \boldsymbol{y}|\mathbf{z} = z, \mathrm{t} = 0)}{q(\mathbf{y} = \boldsymbol{y}|\mathbf{z} = z)}\frac{p(\mathbf{y} = \boldsymbol{y}|\mathrm{t} = 1)}{p(\mathbf{y} = \boldsymbol{y}|\mathrm{t} = 0)}\frac{p(\mathbf{z} = z|\mathrm{t} = 0)}{p(\mathbf{z} = z|\mathrm{t} = 1)}\right] \\
&= \mathbb{E}_{\boldsymbol{y}\sim p(\mathbf{y}|\mathbf{z}=z, \mathrm{t}=0)}\left[\log \frac{\frac{p(\mathbf{y}=\boldsymbol{y}|\mathbf{z}=z, \mathrm{t}=0)p(\mathbf{z}=z|\mathrm{t}=0)}{p(\mathbf{y}=\boldsymbol{y}|\mathrm{t}=0)}}{\frac{p(\mathbf{y}=\boldsymbol{y}|\mathbf{z}=z, \mathrm{t}=1)p(\mathbf{z}=z|\mathrm{t}=1)}{p(\mathbf{y}=\boldsymbol{y}|\mathrm{t}=1)}}\frac{p(\mathbf{y} = \boldsymbol{y}|\mathbf{z} = z, \mathrm{t} = 1)}{q(\mathbf{y} = \boldsymbol{y}|\mathbf{z} = z)}\right] \\
&= \mathbb{E}_{\boldsymbol{y}\sim p(\mathbf{y}|\mathbf{z}=z, \mathrm{t}=0)}\left[\log \frac{p(\mathbf{z} = z|\mathbf{y} = \boldsymbol{y}, \mathrm{t} = 0)}{p(\mathbf{z} = z|\mathbf{y} = \boldsymbol{y}, \mathrm{t} = 1)}\right] + D_{\mathrm{KL}}(p_{\mathbf{y}|\mathbf{z}}^{\mathrm{t}=1}||q_{\mathbf{y}|\mathbf{z}}) \\
&=^* D_{\mathrm{KL}}(p_{\mathbf{y}|\mathbf{z}}^{\mathrm{t}=1}||q_{\mathbf{y}|\mathbf{z}}),
\end{aligned} \tag{49}
$$

where $^*$ uses $p(\mathbf{z}|\mathbf{y}, \mathrm{t} = 0) = p(\mathbf{z}|\mathbf{y}, \mathrm{t} = 1)$.

## B.9 Relation between the different criteria

The different criteria mentioned in section 3 are related by the following expression, which follows from the chain rule of mutual information:

$$
\underbrace{I(\mathbf{e}; \mathbf{z}|\mathbf{y})}_{\text{separation}} + I(\mathbf{e}; \mathbf{y}) = \underbrace{I(\mathbf{e}; \mathbf{y}|\mathbf{z})}_{\text{sufficiency}} + \underbrace{I(\mathbf{e}; \mathbf{z})}_{\text{independence}}
$$

From which we can derive the following:

1. if $\mathbf{y}$ and $\mathbf{e}$ are independent ($I(\mathbf{y}; \mathbf{e}) = 0$) enforcing separation is equivalent to enforcing both sufficiency and independence:

$$
\begin{aligned}
&I(\mathbf{e}; \mathbf{z}|\mathbf{y}) + I(\mathbf{e}; \mathbf{y}) = 0 \\
\iff & I(\mathbf{y}; \mathbf{e}|\mathbf{z}) + I(\mathbf{e}; \mathbf{z}) = 0
\end{aligned} \tag{50}
$$

2. if $\mathbf{y}$ and $\mathbf{e}$ are dependent ($I(\mathbf{e}; \mathbf{y}) > 0$), sufficiency and independence are mutually exclusive conditions (Barocas et al., 2018):

$$
I(\mathbf{e}; \mathbf{y}|\mathbf{z}) + I(\mathbf{e}; \mathbf{z}) \geq I(\mathbf{e}; \mathbf{y}) \tag{51}
$$

# C  Datasets

Here we report the conditional probability distributions used to produce the d-CMNIST and y-CMNIST datasets, underlining the commonalities and differences from the original CMNIST distribution.

Across the three datasets:

- The first two environments are selected for training.

$$p(\mathrm{t} = t | \mathrm{e} = e)$$

|       | $e = 0$ | $e = 1$ | $e = 2$ |
|-------|---------|---------|---------|
| $t = 1$ | 1 | 1 | 0 |
| $t = 0$ | 0 | 0 | 1 |

- The correlation between color c and label y is positive in the first two environments and negative on the last.

$$p(\mathrm{c} = 1 | \mathrm{y} = y, \mathrm{e} = e)$$

|       | $e = 0$ | $e = 1$ | $e = 2$ |
|-------|---------|---------|---------|
| $y = 0$ | 9/10 | 4/5 | 1/10 |
| $y = 1$ | 1/10 | 1/5 | 9/10 |

- For a given color c and digit d, the corresponding picture is a MNIST digit with label d and red (c = 1) or green (c = 0) color. In the discrete settings we assume that pictures contain only color and digit information (ignoring the style) and we construct $\mathbf{x}$ as a concatenation of the color c and digit d: $\mathbf{x} := [\mathrm{c}, \mathrm{d}]$.

## C.1  CMNIST

- All 10 digits d the same probability:

$$\forall d \in 0, \dots, 9 : \ p(\mathrm{d} = d) = 1/10$$

- The 3 environments have the same probability to occur:

$$\forall e \in 0, 1, 2 : \ p(\mathrm{e} = e) = 1/3$$

Note that the first two environments will have probability $0.5$ on the selected training set, while the last one will be drawn with probability $1$ from the test distribution.

- The label $y = 1$ is assigned with probability $0.75$ for digits 5-9 and $0.25$ for digits 0-4 across all environments.

$$p(\mathrm{y} = y | \mathrm{d} = d)$$

|       | $d < 5$ | $d \geq 5$ |
|-------|---------|------------|
| $y = 0$ | 3/4 | 1/4 |
| $y = 1$ | 1/4 | 3/4 |

## C.2  d-CMNIST

- Digits from 0-4 are more likely to occur on the first environment, less likely on the second, while the probability is uniform on the third environment.

$$p(\mathrm{d} \in D | \mathrm{e} = e)$$

|       | $e = 0$ | $e = 1$ | $e = 2$ |
|-------|---------|---------|---------|
| $D = [0, 4]$ | 3/5 | 1/5 | 1/2 |
| $D = [5, 9]$ | 2/5 | 4/5 | 1/2 |

Digits from 0 to 4 (and from 5 to 9) have the same probability:

$$\forall d \in [0,4] : \; p(\mathrm{d} = d) = \frac{\sum_{d' \in [0,4]} p(\mathrm{d} = d')}{5}$$

$$\forall d \in [5,9] : \; p(\mathrm{d} = d) = \frac{\sum_{d' \in [5,9]} p(\mathrm{d} = d')}{5}$$

- The first environment is more likely than the second:

$$p(\mathrm{e} = 0) = 1/2$$
$$p(\mathrm{e} = 1) = 1/6$$
$$p(\mathrm{e} = 2) = 1/3$$

Note that this assignments is designed to ensure that marginally the digits d have uniform distribution on both train and test.

- Labels y are assigned in the same way as the original CMNIST distribution, based on the digits d.

## C.3 y-CMNIST

- Digits 0-4 are more likely to occur for y = 0 and less likely for y = 1:

$$p(\mathrm{d} \in D | \mathrm{y} = y)$$

|  | $y = 0$ | $y = 1$ |
|---|---|---|
| $D = [0,4]$ | 3/4 | 1/4 |
| $D = [5,9]$ | 1/4 | 3/4 |

Similarly to d-CMNIST, digits in the same group ([0,4], [5,9]) have equal likelihood:

$$\forall d \in [0,4] : \; p(\mathrm{d} = d) = \frac{\sum_{d' \in [0,4]} p(\mathrm{d} = d')}{5}$$

$$\forall d \in [5,9] : \; p(\mathrm{d} = d) = \frac{\sum_{d' \in [5,9]} p(\mathrm{d} = d')}{5}$$

Note that the conditional distribution $p(\mathrm{d}|\mathrm{y})$ is the same as the one of the original CMNIST distribution.

- The label $y = 0$ is more likely in the first environment and less on the second. The two labels have the same probability on the third environment:

$$p(\mathrm{y} = y | \mathrm{e} = e)$$

|  | $e = 0$ | $e = 1$ | $e = 2$ |
|---|---|---|---|
| $y = 0$ | 3/5 | 1/5 | 1/2 |
| $y = 1$ | 2/5 | 4/5 | 1/2 |

- The marginal environment distribution is the same as the one described in the d-CMNIST dataset. This assignment ensures that both labels y and digits d have marginal uniform distribution on train t = 1 and test t = 0 splits.

## C.4 Dataset properties

Table 1 reports some of the mutual information measurements for the three CMNIST, d-CMNIST, and y-CMNIST datasets. Note that only representations that contain digit or no information result in zero concept shift.

## C.5 Sampling

Since all the three dataset have uniform digit distribution on both train and test splits ($p(\mathrm{d} = d | \mathrm{t} = 1) = p(\mathrm{d} = d | \mathrm{t} = 0) = 1/10$), the training dataset are produced using the following sampling procedure:

| dataset | Concept shift | | | | Independence | | | Sufficiency | | | | Separation | | |
|---|---|---|---|---|---|---|---|---|---|---|---|---|---|---|
| | $I(y;t\|\mathbf{x})$ | $I(y;t\|c)$ | $I(y;t\|d)$ | $I(y;t)$ | $I_{t=1}(e;\mathbf{x})$ | $I_{t=1}(e;c)$ | $I_{t=1}(e;d)$ | $I_{t=1}(y;e)$ | $I_{t=1}(y;e\|x)$ | $I_{t=1}(y;e\|c)$ | $I_{t=1}(y;e\|d)$ | $I_{t=1}(e;x\|y)$ | $I_{t=1}(e;c\|y)$ | $I_{t=1}(e;d\|y)$ |
| CMNIST | 0.219 | 0.283 | **0.0** | **0.0** | 0.00143 | **0.0** | **0.0** | **0.0** | 0.00854 | 0.00997 | **0.0** | 0.00997 | 0.00997 | **0.0** |
| d-CMNIST | 0.238 | 0.306 | **0.0** | **0.0** | 0.0642 | 0.000636 | 0.0633 | 0.0152 | 0.00588 | 0.0164 | **0.0** | 0.0549 | 0.00760 | 0.0481 |
| y-CMNIST | 0.238 | 0.306 | **0.0** | **0.0** | 0.0331 | 0.0258 | 0.0152 | 0.0633 | 0.0371 | 0.0444 | 0.0481 | 0.00683 | 0.00683 | **0.0** |

Table 1: Mutual information quantities for the three CMNIST datasets. Note that only a representation containing digit or no information results in zero concept shift. Both color and digit satisfy the independence criterion on CMNIST, while digit information is not satisfying sufficiency or separation for the y-CMNIST and d-CMNIST datasets.

1. Determine $p(c, e, y|d, t = 1)$ using Bayes rule:

$$p(c, e, y|d, t = 1) = \frac{p(c, e, y, d|t = 1)}{p(d|t = 1)}.$$

2. Sample a MNIST picture $\tilde{x}$ with label $d$ from the MNIST dataset.

3. Sample from $p(c, e, y|d = d, t = 1)$ to determine color $c$, environment $e$ and label $y$.

4. Color the picture $\tilde{x}$ in green or red depending on the value of $c$ to obtain $x$.

5. Return the triple $(\boldsymbol{x}, y, e)$

Note that in the reported experiments, step 3) is performed at run-time. Therefore, the same picture $\tilde{x}$ can appear in different environments with different colors if sampled multiple times.

Digits used for the train $t = 1$ and test $t = 0$ splits are disjoint, following the traditional train-test MNIST splits. The code used to generate the three datasets can be found at `https://github.com/mfederici/dsit`.

# D  Direct Optimization

In order to optimize for the objective prescribed by the different criteria directly, we define a discrete random variable $\hat{x}$ obtained by concatenating the random variables representing color c and digit d. All the extra information in **x** (e.g. style, position, small rotations, ..) does not interact with the environment e or label y, therefore we can safely assert that **x** is conditionally independent of y and e once $\hat{x}$ is observed:

$$I(\mathbf{x}; ye|\hat{\mathbf{x}}) = 0. \tag{52}$$

At the same time, we can safely assume that color and digit can be identified when a picture **x** is observed:

$$\exists g \ s.t. \ \hat{\mathbf{x}} = g(\mathbf{x}) \implies I(\hat{\mathbf{x}}; ye|\mathbf{x}) = 0. \tag{53}$$

Using the assumptions in 52 and 53, we can infer that $\hat{\mathbf{x}}$ is a sufficient statistic for **x**. In particular, we can show that any (conditional or not) mutual information measurement involving **x**, **y** and **e** is unchanged if $\hat{\mathbf{x}}$ is used instead of **x**.

For example:

$$
\begin{aligned}
I(\mathbf{y};\mathbf{e}|\mathbf{x}) &= I(\mathbf{xy};\mathbf{e}) - I(\mathbf{x};\mathbf{y})\\
&= I(\mathbf{xy};\mathbf{e}) - I(\mathbf{x}\hat{\mathbf{x}};\mathbf{y}) + \underbrace{I(\hat{\mathbf{x}};\mathbf{y}|\mathbf{x})}_{=0 \text{ using eq.53}}\\
&= I(\mathbf{xy};\mathbf{e}) - I(\hat{\mathbf{x}};\mathbf{y}) - \underbrace{I(\mathbf{x};\mathbf{y}|\hat{\mathbf{x}})}_{=0 \text{ using eq.52}}\\
&= I(\mathbf{y};\mathbf{e}) + I(\mathbf{x};\mathbf{e}|\mathbf{y}) - I(\hat{\mathbf{x}};\mathbf{y})\\
&= I(\mathbf{y};\mathbf{e}) + I(\mathbf{x}\hat{\mathbf{x}};\mathbf{e}|\mathbf{y}) - \underbrace{I(\hat{\mathbf{x}};\mathbf{e}|\mathbf{xy})}_{=0 \text{ using eq.53}} - I(\hat{\mathbf{x}};\mathbf{y})\\
&= I(\mathbf{y};\mathbf{e}) + I(\hat{\mathbf{x}};\mathbf{e}|\mathbf{y}) + \underbrace{I(\mathbf{x};\mathbf{e}|\hat{\mathbf{x}}\mathbf{y})}_{=0 \text{ using eq.52}} - I(\hat{\mathbf{x}};\mathbf{y})\\
&= I(\hat{\mathbf{x}}\mathbf{y};\mathbf{e}) - I(\hat{\mathbf{x}};\mathbf{y})\\
&= I(\mathbf{y};\mathbf{e}|\hat{\mathbf{x}}).
\end{aligned}
$$

Note that in this proof we also show $I(\mathbf{x};\mathbf{y}) = I(\hat{\mathbf{x}};\mathbf{y})$ and $I(\mathbf{x};\mathbf{e}|\mathbf{y}) = I(\hat{\mathbf{x}};\mathbf{e}|\mathbf{y})$. The proof for the other reported quantity is analogous.

The dashed trajectories in figure 3 are computed by directly optimizing the objectives reported in section 3 using $\hat{\mathbf{x}}$ instead of $\mathbf{x}$. We use a $[(10 \times 2) \times 64]$ normalized matrix $\boldsymbol{W}$ to parametrize the encoder $q(\mathbf{z}|\mathbf{x})$. Each column of $\boldsymbol{W}$ sums to 1 to represent a valid conditional probability distribution. This matrix maps each combination of digit (10) and color (2) into a representation $\mathbf{z}$ consisting of 64 different options using a dot product:

$$
\mathbf{z} = \boldsymbol{W}^T \text{flatten}(\hat{\mathbf{x}}), \tag{54}
$$

in which flatten($\cdot$) represent the operation flattening the $2 \times 10$ matrix into a 20-dimensional vector. Since $\hat{\mathbf{x}}$ is a categorical variable, this encoder can represents any possible function from $[2 \times 10]$ to the latent space of size $[64]$. We expect the family of encoders parametrized by $\boldsymbol{W}$ to be sufficiently flexible since the dimensionality of the representation is bigger when compared to the number of possible inputs.

All mutual information terms in the objective can be computed and differentiated directly, and we can progressively update $\boldsymbol{W}$ using stochastic gradient descent until convergence. The matrix $\boldsymbol{W}$ is initialized randomly and updated using the ADAM optimizer (Kingma & Ba, 2015) with a learning rate of $10^{-3}$. The optimization stops when the total variation of train and test error is less than $10^{-4}$ for at least 1000 iterations.

The results for different values of the hyper-parameter $\lambda$ are obtained by training a new matrix $\boldsymbol{W}$ from scratch for each considered value of $\lambda$. The parameter $\lambda$ varies from 0 to $10^6$ for models trained using the independence, sufficiency and separation criteria, while a maximum value of 10 is used for the information bottleneck experiments since $\lambda = 2$ is usually sufficient to obtain a constant representation. Further details on the procedure can be found at `https://github.com/mfederici/dsit`.

# E   Training details

Each model in section 4 is composed of the same encoder $q(\mathbf{z}|\mathbf{x})$ and decoder $q(\mathbf{y}|\mathbf{z})$ neural network consisting of multi-layer perceptrons:

- Encoder:

$$
\begin{aligned}
\text{Flatten}() &\to \text{Linear}(\text{in\_size}= 768, \text{out\_size}= 1024) \to\\
&\text{Linear}(\text{in\_size}= 1024, \text{out\_size}= 128) \to\\
&\text{Linear}(\text{in\_size}= 128, \text{out\_size}= 64(\times 2))
\end{aligned}
$$

  with 0.25 dropout probability and ReLU activation in-between each layer. Note that a final size of $128 = 64 + 64$ is used for the VIB experiments to separately parametrize mean and variance of the stochastic encoder $q(\mathbf{z}|\mathbf{x})$.

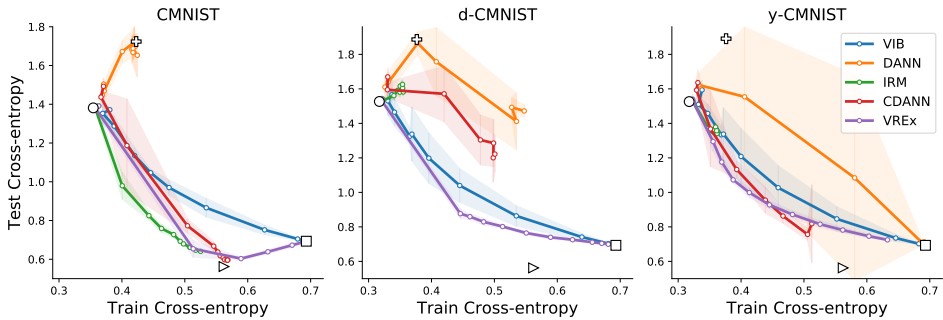

Figure 4: Train and test error on the CMNIST, d-CMNIST and y-CMNIST datasets obtained by training a convlolutional architecture using the same hyper-parameters described in section E. The results are consistent with the ones obtained with a multi-layer perceptron (MLP), which are reported in figure 3.

- Classifier:

$$\text{Linear(in\_size= 64, out\_size= 128)} \rightarrow$$
$$\text{Linear(in\_size= 128, out\_size= 2)}$$

with ReLU activation and softmax output normalization.

Each model is trained for a total of 25000 iterations with batches of size 256 (with the exception of the IRM and VREx model that use a batch size of 4096) using the ADAM optimizer (Kingma & Ba, 2015) with a learning rate of $10^{-4}$. The neural networks are trained for a total of 3000 iterations without any regularization, then $\lambda$ is linearly scaled by a constant value after each iteration ($5 \times 10^{-3}$ for VREx, $2 \times 10^{-2}$ for DANN, CDANN and IRM, $5 \times 10^{-5}$ for VIB). The increment policy has been chosen to maintain stability while exploring the full range of latent representation for each one of the models. The train and test cross-entropy error are measures and reported every 2000 training steps across 3 different seeded runs.

The DANN and CDANN models also require an auxiliary discriminator architecture. In both cases we use multi-layer perceptrons:

$$\text{Linear(in\_size= 64(+2), out\_size= 1024)} \rightarrow$$
$$\text{Linear(in\_size= 1024, out\_size= 128)} \rightarrow$$
$$\text{Linear(in\_size= 128, out\_size= 2)}$$

with ReLU activations, softmax output and spectral normalization (Miyato et al., 2018) for additional stability. The $+2$ in parenthesis accounts for the extra dimensions that are required for the label concatenation in the CDANN model. In between each update of the encoder and classifier neural network, the discriminator model is trained for a total of 64 iterations with a learning rate of $10^{-4}$. This procedure is required to ensure stability of the adversarial training. The entirety of the experiments reported in this work required approximately a total of 250 compute hours on Nvidia 1080 GPUs. The code used to train the architectures is publicly available at `https://www.github.com/mfederici/dsit`.

### E.1 CNN results

In order to validate the result obtained with the different regularization strategies, we perform some additional experiments using a convolutional encoder architecture:

$$\text{Conv(in\_channels= 2, out\_channels= 32, kernel\_size= 5, stride= 2)} \rightarrow$$
$$\text{Conv(in\_channels= 32, out\_channels= 64, kernel\_size= 5, stride= 2)} \rightarrow$$
$$\text{Conv(in\_channels= 64, out\_channels= 128, kernel\_size= 4, stride= 2)} \rightarrow$$
$$\text{Flatten()} \rightarrow \text{Linear(in\_size= 128, out\_size= 64)}$$

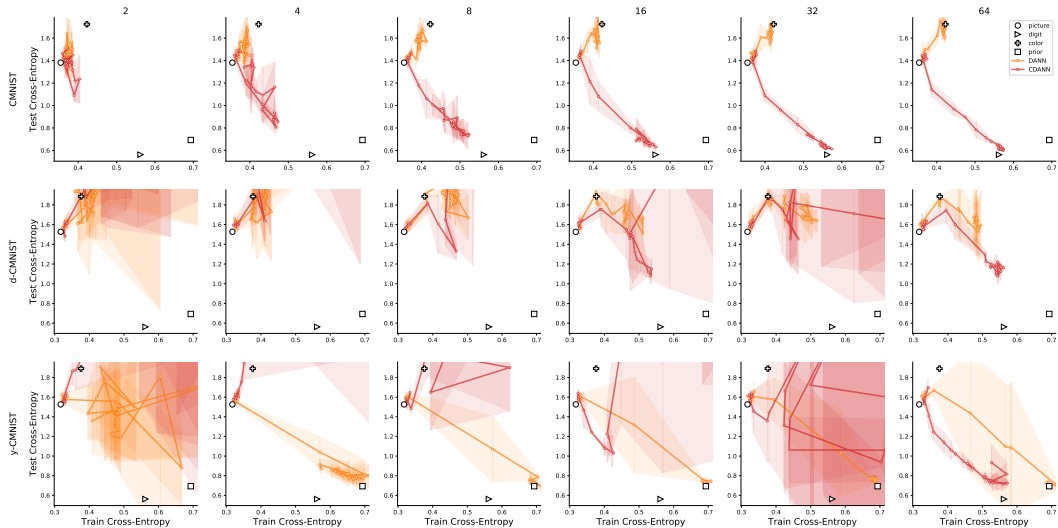

Figure 5: Measurements of train and test cross-entropy for varying number of discriminator training iterations (columns) on the three CMNIST variations (rows) on three different runs. The model convergence and stability is strongly affected by the number of discriminator iterations, especially on the d-CMNIST and y-CMNIST datasets.

with ReLU activations and dropout.

The corresponding results reported in Figure 4 for the three CMNIST variations are consistent with the ones reported in figure 3, underlying that the observed behaviour depends on the model objectives more than the chosen architectures.

## F    Hyper-parameters and their effect

### F.1    On the number of adversarial steps

The results reported in figure 5 clearly show that the effectiveness of the DANN and CDANN model strongly depends on the number of adversarial iterations used for each training step of the encoder model. The plots in the figure 5 display that an increased number of adversarial iterations helps with both stability and performance of the model. This indicates that the optimality of the adversarial predictors $q(\mathbf{e}|\mathbf{z})$ and $q(\mathbf{e}|\mathbf{z}, \mathbf{y})$ plays a fundamental role in the training procedure of the model.

Note that the training cost and time scales up linearly with the number of adversarial iterations. This becomes especially problematic with bigger models and datasets.

### F.2    On the size of the training batch

We observed that both the IRM and VREx models are sensitive to the choice of the size of the training batch used for optimization. We hypothesize that this is due to the fact that both method rely on the computation of a higher order batch statistic (variance and gradient norm).

Figure 6 reports train and test cross-entropy obtained for different values of batch size. We observe that increasing batch size is required to stabilize the training procedure and improve generalization results at the cost of increased memory and computational requirements.

### F.3    Practical limitations of Invariant Risk Minimization

We observed that the IRM model is unable to minimize the test error on the d-CMNIST dataset despite the optimality of the sufficiency criterion. We identify two main reasons that can explain this issue:

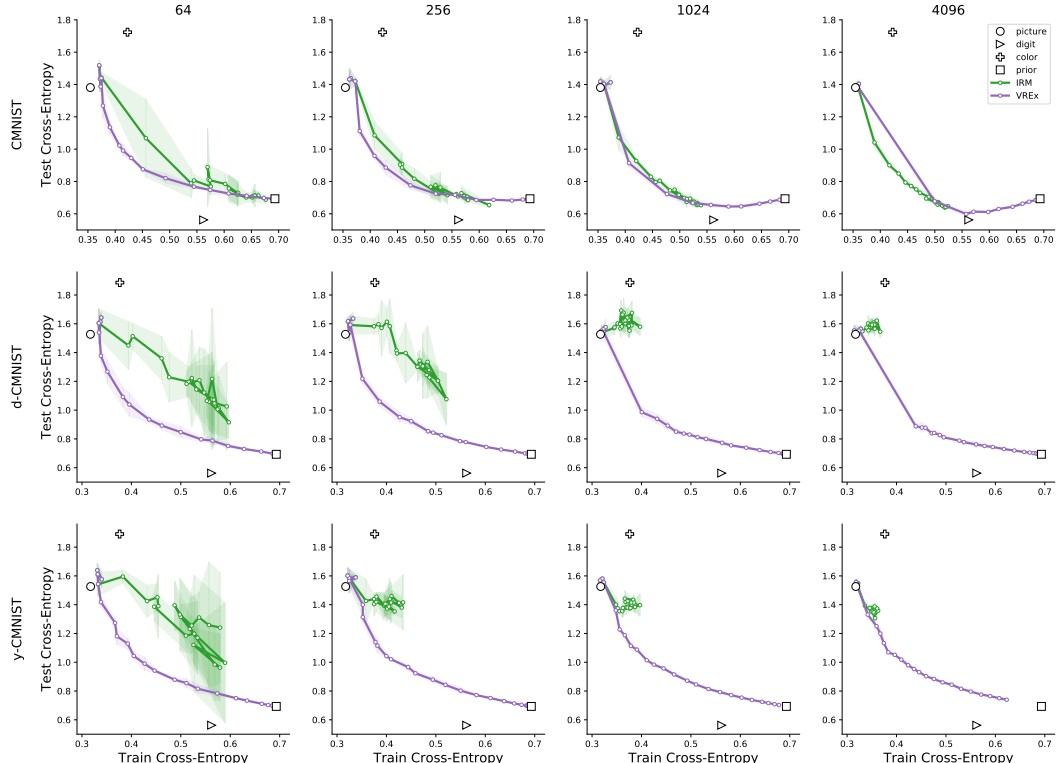

Figure 6: Visualization of the effect of different choices of batch size (different columns) for the IRM and VREx models on the three CMNIST variations (rows) using the MLP architectures described in appendix E. Increasing the batch size generally results in better stability and performance at the cost of extra computation time. The models in the pictures are trained for the same amount of iterations.

1. The Invariant Risk Minimization introduces a relaxation of the sufficiency criterion:

$$\text{Sufficiency} \implies \text{IRM-optimality}, \quad \text{IRM-optimality} \not\implies \text{Sufficiency}.$$

As a consequence, the model can converge to an IRM-optimal solution which is not enforcing sufficiency. Further discussion on the relation between sufficiency and IRM is discussed in Huszár (2019).

2. The gradient penalized by the regularization term $\mathbb{E}_{\boldsymbol{e}}||\nabla \ell_{\boldsymbol{e}}(q_{\mathbf{y}|\mathbf{z}} \circ q_{\mathbf{z}|\mathbf{x}})||^2$ becomes too small to produce meaningful encoder updates.

Figure 7 shows that the gradient on the different datasets quickly goes to zero as the regularization strength $\lambda$ is increased. Increasing the batch size results in a decrease in the variance of the gradient estimate, but it does not increase its magnitude. This suggests that either the model already achieved IRM-optimality, or that the gradient is so small that the variance of the regularization term needs to be decreased by orders of magnitudes to result in meaningful updates. Although possible, the latter option would imply that drastically bigger batch size are needed, reducing the applicability of the IRM model to larger settings.

## G  Generalizing constraints from the train distribution

In section 3, we mention that different criteria implicitly rely on the assumption that enforcing a constraint (or maximizing predictive information) on the training distribution is sufficient to ensure that the constraint holds for the whole distribution, even outside of the selection. Here we report a simple example in which these assumption are clearly violated to underline the importance of environment variety at training time and pinpoint the limitations of the approaches explored in this work.

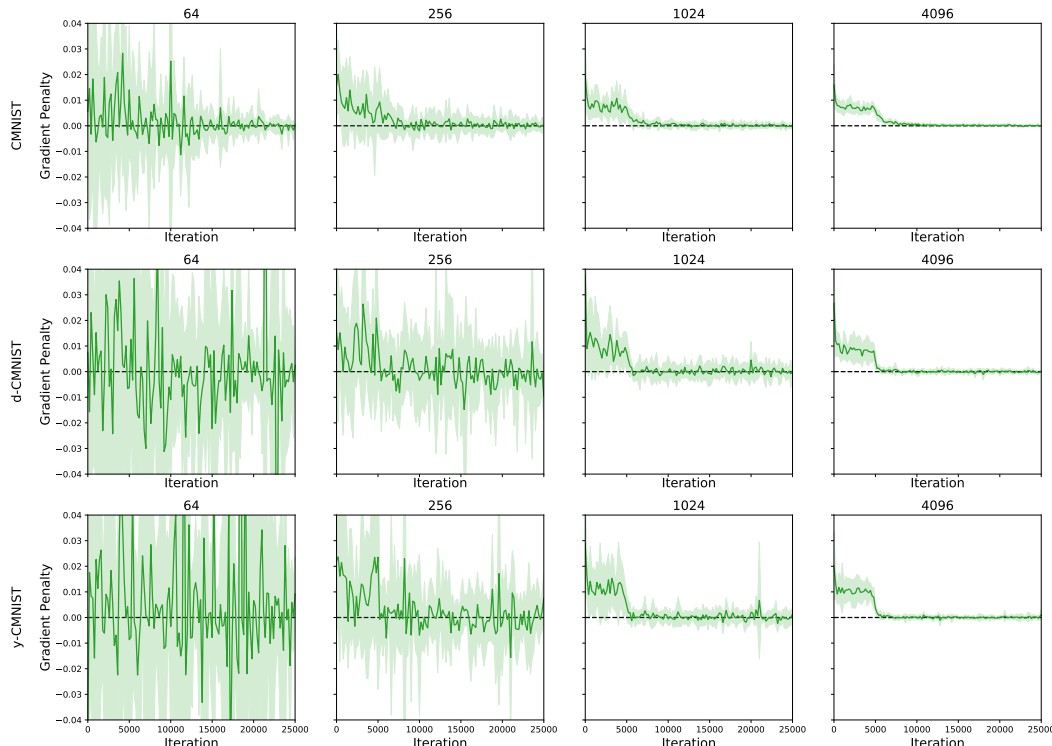

Figure 7: Gradient penalty measured during training of the IRM models using different batch size on the three variations of the CMNIST dataset. Increasing the size of the training batches reduces the variance of the gradient but it does not increase its magnitude.

Consider the problem of constructing a classifier to distinguish pictures of cows from camels. To simplify our reasoning, we will assume that each picture contains only animal features $a \in$ {patches, horns, humps, ..} and background features $b \in$ {sand, grass}. The data is collected from different locations $\mathbf{e}$, which represent the environmental factors. If the training distribution does not contain any example of cows on sandy terrains or camels on pastures, we can not expect any of the described criteria to successfully classify pictures. This happens for two main reasons:

1. On the training distribution animal and background feature are equally predictive of the label:

$$I_{t=1}(y; a) = I_{t=1}(y; b) = I_{t=1}(y; ab).$$

   As a consequence, we can not expect the encoder $q(\mathbf{z}|\mathbf{x})$ to preferentially extract animal features $a$, especially when the background $b$ is much easier to identify.

   Note that the amount of predictive information for the background $b$ on the distribution that has not been selected (camels on pastures, cows on sand for $t = 0$) is strictly less than the amount of information carried by $a$:

$$I_{t=0}(y; b) < I_{t=0}(y; a) = I_{t=0}(y; ab).$$

   This is because not all camels are necessarily in sandy locations, but all of them have humps $(H_{t=0}(y|b) > H(y|a) = 0)$. In other words, the lack of diversity in the selection makes the background information $b$ as predictive as the animal features $a$ on the training distribution, although background is not more predictive than the animal features in general.

2. Background information is stable across all the training locations $\mathbf{e}$:

$$p(y, b|e, t = 1) = p(y, b|t = 1).$$

   This happens since, within the training locations, all cows are on green patches and all camels are on sandy terrains. Since the joint distribution of background and label is the same across different training locations, the constraint imposed by both sufficiency and separation

criteria cannot remove the background information b from the representation $\mathbf{z}$. Since both $p(\mathbf{y}|\mathbf{b}, \mathbf{t}=1)$ and $p(\mathbf{b}|\mathbf{y}, \mathbf{t}=1)$ appear stable we have:

$$I_{\mathbf{t}=1}(\mathbf{y}; \mathbf{e}|\mathbf{b}) = 0, I_{\mathbf{t}=1}(\mathbf{e}; \mathbf{b}|\mathbf{y}) = 0.$$

In other words, no regularization strength $\lambda$ can force a model trained using the sufficiency or separation (or independence) criterion to remove background information from the representation.

Note that even if background appears to be a stable feature on the selection, it is not stable in general and we have:

$$I(\mathbf{y}; \mathbf{e}|\mathbf{b}) > 0, I(\mathbf{e}; \mathbf{b}|\mathbf{y}) > 0,$$

since both $p(\mathbf{y}|\mathbf{b})$ and $p(\mathbf{b}|\mathbf{y})$ can change for some environmental conditions that have not been selected. This is because in some (unobserved) locations it is possible to find cows on sandy beaches or camels on grasslands.

To summarize, lack of diversity in the training environment may compromise the effectiveness of the criteria mentioned in this analysis. This is because models can rely on features that appear to be stable on the training distribution to make predictions. Whenever the stability is due to lack of diversity in the data selection, we cannot expect optimal out-of-distribution performance.