# OpenReview forum: "An Information-theoretic Approach to Distribution Shifts"
_NeurIPS.cc/2021/Conference — NeurIPS 2021 Poster_

### Official Review · Reviewer_S3bF · 2021-07-07

**Rating:** 5
**Confidence:** 1

**Summary:**

The paper studies the distributional shift setting, in which test and training distribution are different.  The distribution shift is described in terms of mutual information of the data with a binary selection variable $t$ (train or test).
Several propositions describe relations between the distribution of test and train covariates and labels and a latent space.
Based on those observations, several regularization criteria are suggested.
The experiments compare various settings of distributional shifts and popular methods.

**Limitations And Societal Impact:**

The limitations are discussed in the main review. There is no potential negative societal impact.

**Main Review:**

Strengths:
- Distributional Shift is an interesting and important problem.
- The information theory approach gives a novel and intuitive description of the problem.
- The experiments test different settings of distributional shift and compare the performance of popular approaches.

*******************

Weakness:
- I think the paper lacks significant technical results. I may have missed some insights, but I think that the propositions don't lead to significant and quantifiable conclusions about the distributional shift problem.
The methods tested in the experiments section are claimed to be a relaxation of the criteria derived from the framework. This can be a very interesting conclusion, but it is not studied thoroughly in the paper (Sect 5.1 doesn’t elaborate the description of the methods.)

- There is also no mention of the finite-sample nature of learning problems. The conclusion from Prop 1. about the maximum-likelihood solution seems unfounded if there is no mention of the data used for the estimation.

- The objective function in Sect 2.5 resembles Variational Autoencoders, but it is not discussed in the paper.

- Section 2.3 -  Lines 105-106 - “Whenever the JSD is small then the latent OOD error is small” - this doesn’t seem to follow from (8) since the JSD is a lower bound.


*******************
Clarity:
The paper is generally well written.
However, the introduction is not informative enough about the contributions of the paper.
The different sections feel disjointed with no clear organizing structure.
For example, section 2 is titled “Framework and Problem Statement” but contains propositions. It is not clear where the main results of the paper start.

*******************
Minor comments:
- Line 98 - the sentence is not clear.
- Line 191 - typo - “t”
- It seems to be a problem with the PDF, Fig 3 is missing when viewing the file with Adobe Reader (but it is fine with other software).



**Time Spent Reviewing:**

6

---

> ### Author Response · Authors · 2021-08-10
> **Response to Reviewer S3bF**
>
> ## Answers and Comments
> We thank the reviewer for their insightful comments and suggestions for clarity enhancement. Our comments are listed below:
>
> 1) **"I think the paper lacks significant technical results. I may have missed some insights, but I think that the propositions don't lead to significant and quantifiable conclusions about the distributional shift problem."**
>
> We summarize our main contributions in the general comment to the reviewers. We believe that our analysis can be useful to define new directions of research and better understand the issues of recent approaches to the distribution shift problem.
>
> 2) **"The methods tested in the experiments section are claimed to be a relaxation of the criteria derived from the framework. This can be a very interesting conclusion, but it is not studied thoroughly in the paper (Sect 5.1 doesn’t elaborate the description of the methods.)"**
>
> The independence, separation and, sufficiency criteria are driving the design of the losses for the models reported in the analysis (as reported in the respective papers).
>
> Adversarial training can be used to approximately minimize and control dependencies with an environment variable [1].
> This principle clearly connects the DANN and CDANN to the independence and separation criteria respectively.
> A discussion regarding the IRM model and the sufficiency criterion is reported in [2], while answer (1.) to Reviewer y2kJ aims to clarify our claims about the VREx model.
>
> The main text will be updated to clarify our claims.
>
> 3) **"There is also no mention of the finite-sample nature of learning problems. The conclusion from Prop 1. about the maximum-likelihood solution seems unfounded if there is no mention of the data used for the estimation."**
>
> Even if the issue of fitting a model to an unknown distribution given a finite amount of samples is extremely relevant in the context of deep learning, this problem can be considered complementary to the issues caused by distribution shifts that are analyzed in this work.
>
> Our analysis focuses on the situation in which we have enough samples to have a good model $q(y|{\bf x})$ of the training distribution $p(y|{\bf x}, t=1)$, and our goal is to show that, even in this ideal setting, we are still bound to have a large test error due to the selection bias and analyze the distribution shift problem from an information-theoretic perspective.
>
> 4) **"The objective function in Sect 2.5 resembles Variational Autoencoders, but it is not discussed in the paper."**
>
> The paper includes a discussion regarding the Variational Information Bottleneck model (VIB) of which Variational Autoencoders can be seen as a specific instance [3] (Appendix B). This model (and design principle) will be analyzed in more detail in the main text with a dedicated part in section 3 (answer (6.) to Reviewer rngK) .
>
> 5) **"Section 2.3 - Lines 105-106 - “Whenever the JSD is small then the latent OOD error is small” - this doesn’t seem to follow from (8) since the JSD is a lower bound."**
>
> We justify the statement in lines 105-106 with the following statements:
>
> a) Since both KL-divergence and JSD are measures of divergence, for $D_{JSD}(p_{y|{\bf x}}||q_{y|{\bf x}})=0$ we have $D_{KL}(p_{y|{\bf x}}||q_{y|{\bf x}})=0$
>
> b) Since both Jensen-Shanon divergence and Kullback-Leibler divergence are instances of $f$-divergeces, we can write an upper bound for the OOD error as a function of the Jensen-Shannon distance using Pinsker-type inequalities (Theorem 3 in [4], Theorem 1 in [5]):
> $$
> \sqrt{D_{JSD}(p_{y|{\bf x}}^{t=0}||q_{y|{\bf x}})}\ge \sqrt{2}\left(\frac{m({\bf x})\log m({\bf x})}{1-m({\bf x})}+\frac{M({\bf x})\log M({\bf x})}{1-M({\bf x})}\right)D_{KL}(p_{y|{\bf x}}^{t=0}||q_{y|{\bf x}})
> $$
> with
> $
> m({\bf x}) = \min_y\frac{p(y|{\bf x},t=0)}{q(y|{\bf x})},\ \  M({\bf x}) = \max_y\frac{p(y|{\bf x},t=0)}{q(y|{\bf x})}
> $
>
> Therefore whenever the support of $q(y|{\bf x})$ covers the support of $p(y|{\bf x}, t=0)$, the OOD error is finite and bounded by the previous expression.
>
> The detailed derivation will be included in the appendix.
>
> ## Clarity
>
> To improve the clarity we will :
> - Update the introduction to make sure the contributions are immediately clear to the reader
> - Update section 2 to better distinguish between problem statement and main results
> - Rephrase line 98
> - Update the sections to be more cohesive.
>
> We will make sure that the PDF can be visualized with all the common software.
>
> ## References:
> *[1] Xie, Qizhe, et al. "Controllable invariance through adversarial feature learning." arXiv preprint arXiv:1705.11122 (2017).*
>
> *[2] Huszár, Ferenc. “Invariant Risk Minimization: An Information Theoretic View.” 19 July 2019, www.inference.vc/invariant-risk-minimization/.*
>
> *[3]Alemi, Alexander A., et al. "Deep variational information bottleneck." arXiv preprint arXiv:1612.00410 (2016).*
>
> *[4]: G. L. Gilardoni, "On Pinsker's and Vajda's Type Inequalities for Csiszár's $f$ -Divergences," in IEEE Transactions on Information Theory, vol. 56, no. 11, pp. 5377-5386, Nov. 2010, doi: 10.1109/TIT.2010.2068710.*
>
> *[5]: Binette, Olivier. "A note on reverse Pinsker inequalities." IEEE Transactions on Information Theory 65.7 (2019): 4094-4096.*

---

### Official Review · Reviewer_rgnK · 2021-07-15

**Rating:** 7
**Confidence:** 3

**Summary:**

The authors describe the problem of data shift from a information
theoretic perspective. They identify different sources of error and compare
the effect of different regularization objectives on creatively derived datasets
in a controlled setting which cannot be achieved in most normal datasets.

**Limitations And Societal Impact:**

Yes, they have adequately addressed the limitations in the appendix. I am not aware of any negative societal impact.

**Main Review:**

## Pros

- the description of the problem of distributional shift as being a result of
  not truly IID sampling the real distribution is informative and intuitive
- The construction of the datasets is sound and captures complex interactions between different dataset selection biases.
- The authors are very thorough in their theoretical framework which makes a compelling argument for analyzing future OOD-robust models under this controlled setting.

# Cons

- The colors in figure three are too hard to distinguish from one another. The
legends are also too small for being able to make out what they are showing at
a glance.

- I find the naming of the criteria in sections 3.1, 3.2, and 3.3 to be quite
confusing. Why is 3.1 called "Independence Criterion" when actually all of the
sections 3.1, 3.2, 3.3 are encouraging some form of independence? I realize
that these are cited as coming from another source, but I still think they
should be clarified so that readers do not become confused.

- It would be good to include some qualitative samples from the CMNIST dataset
in order to gain more intuition about what is being said in the paper.

- Why are none of the models evaluated on cross-criteria? The models seem to be
chosen for the specific inductive biases which predispose them to do better on
certain criteria. If this is the case, then it would be very interesting to see
how they perform on the other criteria, in order to validate that the inductive
biases are correct.

- I think it would be interesting to analyze some popular convolutional models
such as resnets in the controlled setting to see how their performance breaks
down (like equation 3) and then verify if the trend continue on larger datasets
such as CIFAR-10/100 with an OOD dataset for testing. If the best model in the
controlled setting also performed the best in this experiment (on some other standard OOD metrics which can be measured), that would be
extremely strong validation of the main theoretical findings.

## Questions

- In Figure 1 it is not explicitly clear how there is sample bias in the data
$p(y | x_1)$. It seems as if there was actually no selection bias, but just
another feature was added. Is there something I am missing?

- The anonymous code repo in the appendix appears to be incomplete, as I cannot
see the actual training script anywhere in these files. Is there something I am
missing?

- Equation 48 in the appendix seems to show that the encoder is only a linear
function. Is there really no non-linearity in that function?

- Equation 48 says it was used to compute the latent representations for $z$ in
figure 3, but then in section E of the appendix, the authors state that the
encoder has 3 hidden layers. Which one is correct?

- What is the "bottleneck" criterion in figure 3 and how does this relate to
the criteria highlighted in sections 3.1 - 3.3?

# Minor

- L93 I find this confusing because the equations show "information loss" and
  then "latent error," but in the text L93 the authors refer to them in the
  opposite order and then on L94-95 they refer to the "former and latter." I
  think the former and latter should match the order of the terms in the
  equations in order to avoid confusion.

Overall, I am extremely positive on the paper and the main theoretical
motivations and findings. However, I feel there are further experiments which could be
done to further validate the findings (the last two bullet points of the **Cons**
review). I feel the organization and clarity of each section could also use improvement, as I had to read each section multiple times in order to grasp what was being said.

**Time Spent Reviewing:**

5-6 hours

---

> ### Author Response · Authors · 2021-08-10
> **Response to reviewer rngK**
>
> ## Answers and Comments
> We thank the reviewer for the insightful comments and questions that are addressed in this section:
>
> 1) **"Why are none of the models evaluated on cross-criteria? The models seem to be chosen for the specific inductive biases which predispose them to do better on certain criteria. If this is the case, then it would be very interesting to see how they perform on the other criteria, in order to validate that the inductive biases are correct."**
>
> The datasets in our experiments are designed to cover all the possible feature stabilities ($p(y,{\bf z})$ for CMNIST, $p(y|{\bf z})$ for d-CMNIST,  and $p({\bf z}|y)$ for y-CMNIST).
> Each criterion (dashed lines in Figure 3) and implementation (solid lines) is tested in all the scenarios.
>
> We underline that criteria (Independence, Sufficiency, Separation) and implementations (DANN, CDANN, VREx, IRM, VIB) that look for the "wrong stability" do not manage to achieve optimality in section 5 (lines 292-322).
> We apologize for possible confusion that might have arisen from the use of the ambiguous word "model" to refer to the aforementioned implementations in the experimental section. This will be addressed in our edit.
>
> 2) **"I think it would be interesting to analyze some popular convolutional models such as resnets in the controlled setting to see how their performance breaks down (like equation 3) and then verify if the trend continue on larger datasets such as CIFAR-10/100 with an OOD dataset for testing. If the best model in the controlled setting also performed the best in this experiment (on some other standard OOD metrics which can be measured), that would be extremely strong validation of the main theoretical findings."**
>
> The main goal of this work is to discuss and analyze the design principle behind the different models and better understand the source of error when there is a shift between train and test distribution.
> Any quantitative empirical evaluation of these criteria would require a reliable and differentiable estimation of the information-theoretical quantities in analysis. This is possible on the CMNIST dataset since we have full knowledge of the simple data-generating process (answer 3 reviwer y2kJ).
>
> More complex datasets (for which the data-generating process is not fully known) would require sample-based mutual information estimation/optimization. Unfortunately, the existing mutual information estimators are not good enough to accurately estimate/optimize for the quantities of interest ([1], [2]). Furthermore, the effect of the complementary problem of under/over-fitting (which is not the focus of this work, as explained in answer 3 to Reviewer S3bF) would become more relevant on bigger datasets.
>
> We believe that the value of this paper lies in the analysis of the OOD error and its relation to the different criteria, therefore we focus our experimental section on the simplest possible settings in which we can clearly show the shortcomings of the design criteria and models.
>
> 3) **"In Figure 1 it is not explicitly clear how there is sample bias in the data . It seems as if there was actually no selection bias, but just another feature was added. Is there something I am missing?"**
>
> The sample bias in figure 1 is due to the fact that the majority of the people (for which the values of blood pressure and cholesterol levels are known) have reported chest pain. This variable represents the *"environmental variable"* $e$ on which the selection $t$ is based on.
> The "pain"/"no-pain" feature is considered to be not observed during test time, and the whole picture is meant to introduce a simplified example of a real-world scenario in which selection bias causes some predictive error, which can potentially be corrected using additional information regarding the selection.
>
> We will clarify this assumption and the role of the chest-pain variable in the figure caption.
>
> 4) **The anonymous code repo in the appendix appears to be incomplete, as I cannot see the actual training script anywhere in these files. Is there something I am missing?**
>
> We did not include the code for the neural network models and training since our implementation is based on some publicly available resources [3] and the hyper-parameters used for neural network architectures and training are specified in the appendix.
>
> Our repository currently includes all the code for mutual information estimation and optimization (to reproduce the results reported with the dashed lines in figure 3) and implementation of the CMNIST, d-CMNIST, and y-CMNIST datasets.
> We will include the model definition and training as suggested to have a self-contained repository.
>
> 5) **"Equation 48 in the appendix seems to show that the encoder is only a linear function. Is there really no non-linearity in that function?" "Equation 48 says it was used to compute the latent representations for  in figure 3, but then in section E of the appendix, the authors state that the encoder has 3 hidden layers. Which one is correct?"**
>
> Our experimental section (and figure 3) includes results obtained in two different ways:
>
> a) When optimizing the different criteria directly (bottleneck, sufficiency, separation, independence; dashed lines in figure 3) we encode a discrete sufficient statistic $\bf x'$ of the pictures $\bf x$ into a discrete categorical representation $\bf z$ (the code is available in the anonymous repository). In this setting, the family of all possible encoding distribution can be represented with a matrix (linear encoding). Note that since the dimensionality of $\bf z$ is higher the number of possible values of $\bf x'$ ($64=|\mathcal{Z}|>|\mathcal{X’}|=20$) we are not imposing any particular restriction or bias in the representation.
>
> b) When training the models (VIB, IRM, VREx, DANN, and CDANN; solid lines in figure 3) we use the pictures ${\bf x}$ as an input and a flexible neural network encoder to model $q({\bf z}|{\bf x})$ since a linear encoder would impose additional (unwanted) restrictions on the representation.
>
> 6) **"What is the "bottleneck" criterion in figure 3 and how does this relate to the criteria highlighted in sections 3.1 - 3.3?"**
>
> The *"bottleneck criterion"* refers to the Information Bottleneck criterion [4], which relates to the idea that one can simply discard information to improve test generalization (i.e. $\mathcal{R}(q_{z|x})=I({\bf x};{\bf z})$). Although the Information Bottleneck criterion is not generally used in the context of distribution shifts, we will add a corresponding part in section 3 to better discuss the rationale behind this approach and clarify our analysis.
>
> ## Clarity
>
> - The colors in Figure 3 and the legend size will be updated as requested.
> - Each subsection of section 3 will include a sentence to clarify the intuition behind the naming convention.
> - The text at line 93 will be updated as recommended.
> - We also believe that creating a dedicated section to the bottleneck method and clarifying the difference between the experimental setups as suggested will improve the readability of the paper.
>
> ## References:
>
> *[1] McAllester, David, and Karl Stratos. "Formal limitations on the measurement of mutual information." International Conference on Artificial Intelligence and Statistics. PMLR, 2020.*
>
> *[2] Poole, Ben, et al. "On variational bounds of mutual information." International Conference on Machine Learning. PMLR, 2019.*
>
> *[3] DomainBed: https://github.com/facebookresearch/DomainBed*
>
> *[4] Tishby, Naftali, Fernando C. Pereira, and William Bialek. "The information bottleneck method." arXiv preprint physics/0004057 (2000).*

---

> > ### Comment · Reviewer_rgnK · 2021-08-15
> > **Score Updated**
> >
> > Thank you for your responses, I have updated my score. The main reason behind my initial score was a similar concern as reviewer S3bF which was that the work doesn't lead to any quantifiable and 'actionable' conclusions about the problem of distributional shifts. However, I think this work does a good job of exploring different sources of possible error and I actually think there should be more works like this which focus on asking the proper questions even if there is no actionable result in the end.
> >
> > I think this work will likely be used in future approaches to further investigation or approaches to solving the problem of distributional shifts.

---

### Official Review · Reviewer_ukrp · 2021-07-19

**Rating:** 7
**Confidence:** 3

**Summary:**

This paper investigates distribution shift from an information-theoretic perspective. It outlines a theoretical framework for describing different sources of modeling error, including that due to distribution shift, in the presence of a latent representation. It uses the framework to categorise existing approaches to reducing distribution shift according to the type of regularization criteria they use. It explains the assumptions underlying the success of these approaches, and demonstrates empirically, on synthetic data, that when these assumptions fail to hold, performance drops.

**Limitations And Societal Impact:**

Technical limitations are briefly commented on. No potential negative societal impact is discussed.

**Main Review:**

**Originality**

I’m not an expert in this area, but to the best of my knowledge the theoretical presentation is original, as is the demonstrated experiments. It appropriately cites the selected approaches, which are investigated experimentally. I would have liked to see references given in section 3.3 on the separation criteria, as in 3.1, 3.2 (this is given in related work, but good to have it here too in my opinion).

**Quality**

The quality of the theory is very good. Ideally I would have liked to see the experiments section pushed further, in particular on the neural network approaches. Only one type of architecture for all methods was investigated, and some surprising results were found (VREx performing worse than expected on CMNIST and d-CMNIST; CDANN performing worse than expected on y-CMNIST), which don’t line up well with the theory presented. It’s not really possible with the current experiments to tell whether this is something fundamental about the method, or if it’s something about the architecture, optimizer or other hyperparameters.

**Clarity**

The paper is well-written overall.
- One minor suggestion is to move the related work section to either after the introduction, or just before the conclusion (perhaps the latter works better considering the context of the paper is helpful to understand the related work section). In its current position it breaks the flow between theory and experiments.
- I would also prefer to call the ‘OOD’ terms ‘test’ instead, as this is more standard, and perhaps more correct (just because a sample is not in the training set, doesn’t mean it is out of distribution in some fundamental way - although sometimes this is the case).
- I found equation 9 and its preceding paragraph hard to follow, consider rewriting, and giving an explanation for the derivation of 9.
- Line 141: doesn’t lambda also trade off aspects of (i), as well as between (ii) and (iii)?
-Line 171: suggest rewrite I(e;y|z) as I(y;e|z) for consistency with eq 11 - even though Mutual Info is symmetric, it will be less confusing to reader.

**Significance**
The main limit on significance is the use of synthetic datasets - not just because this gives us limited experimental evidence about the effectiveness of methods for distribution shift, but because the theoretical terms would be challenging to estimate in real-world data (the paper acknowledges this shortcoming). Whether the presented analysis is actually useful would have to be shown in follow-up work.


**Time Spent Reviewing:**

6

---

> ### Author Response · Authors · 2021-08-10
> **Response to reviewer ukrp**
>
> ## Answers and Comments
> We thank the reviewer for the relevant questions and suggestions.
> We address their comments in this section:
>
> 1) **"Only one type of architecture for all methods was investigated, and some surprising results were found (VREx performing worse than expected on CMNIST and d-CMNIST; CDANN performing worse than expected on y-CMNIST), which don’t line up well with the theory presented. It’s not really possible with the current experiments to tell whether this is something fundamental about the method, or if it’s something about the architecture, optimizer or other hyperparameters."**
>
> We understand the reviewer's concerns regarding the experimental section.
> The architectures and optimizations have been fixed in advance so that the architectures are flexible enough to model the true distributions but small enough to limit the relevance of regularization strategies.
> Since we cannot exclude the option that, by a mere chance, we might have favored one of the models, we will include results obtained with smaller and bigger architectures in the appendix.
> Some of the reasons behind the discrepancy between theoretical results and the ones obtained with the analyzed models are discussed in appendix E.1 and E.2.
>
> 2) **"Line 141: doesn’t lambda also trade off aspects of (i), as well as between (ii) and (iii)?"**
>
> Re-writing expression (10), we can show that the only term that depends on the latent classifier $q({\bf y}|{\bf z})$ is $D_{KL}(p_{{\bf y}|{\bf z}}^{t=1}||q_{{\bf y}|{\bf z}})$. Therefore, for a sufficiently flexible family of latent classifiers, the minimization problems for $q({\bf z}|{\bf x})$ and $q({\bf y}|{\bf z})$ can be seen as independent.
>
> We will clarify this additional assumption with a footnote in the main text.
>
> ## Clarity
> We summarize the clarity enhancements that will be applied to the paper according to the reported suggestions:
>
> - We move the related work section after the experimental one.
>
> - Rename 'OOD' error to 'test' error
>
> - Rephrase lines 126-131 (before equation 9) to improve clarity and readability
>
> - Re-write all the mutual information terms using the same ordering for consistency
>
> - Add citations to section 3.3

---

### Official Review · Reviewer_y2kJ · 2021-07-20

**Rating:** 6
**Confidence:** 4

**Summary:**

This paper motivates a series of information based analysis and methods. The authors first propose an information theoretic analysis framework for out-of-distribution generalization problem. Some corresponding generalization bounds are brought forward. Three different kinds of regulators are further studied. The effectiveness of proposed methods are demonstrated through numerical experiments.

**Limitations And Societal Impact:**

The reviewer doesn't foresee a limitation on societal impact.

**Main Review:**

The writing of the main paper and appendix are clear.  The reviewer lists some concerns here.

1. Why the authors claim that VREx is a relaxation of sufficiency criterion? VREx requires the stability of joint distribution of $(y, z)$ to achieve a stable loss value across domains.

2. The ablation about combination of different criterions is lacked, which would be an interesting direction.

3. How to numerically estimate the mutual information?

**Time Spent Reviewing:**

5 hours

---

> ### Author Response · Authors · 2021-08-10
> **Response to Reviewer y2kJ**
>
> ## Answers and Comments
> We thank the reviewer for the relevant questions and suggestions for future research.
> Here we address their questions and concerns:
>
> 1) **"Why the authors claim that VREx is a relaxation of sufficiency criterion? VREx requires the stability of joint distribution of $(y,z)$ to achieve a stable loss value across domains."**
>
> We agree with the reviewer in the claim that VREx addresses the stability of the features $p({\bf z})$ as well as the stability of the predictive distribution $p(y|{\bf z})$.
> Although the sufficiency criterion is one of the design principles for the VREx model, we will update the main text to clarify that the VREx objective can be seen as an instance of the aforementioned criterion only in the absence of covariate shift (as in the CMNIST dataset).
>
> In this setting, clearly, sufficiency $I({\bf y};{\bf e}|{\bf z})=0$ implies stability of the joint distribution $p({\bf y},{\bf z})$ (since $p({\bf z})$ is stable), and, therefore, the same error will be registered across environments.
>
> On the other hand, having zero risk variance is not a sufficient condition to guarantee that sufficiency is enforced.
> In facts, a model $q(y|{\bf z})$ that consistently underestimates the target $y$ for some environment $e_1$ and overestimates it by the same amount on some other environment $e_2$ would have zero error variance even if $I({\bf y};{\bf e}|{\bf z})>0$ (since $p(y|{\bf z}, e_1)\neq p(y|{\bf z}, e_2)$).
>
> Further details regarding the relation between VREx and the mentioned criteria will be included in the appendix.
>
> 2) **"The ablation about combination of different criterions is lacked, which would be an interesting direction."**
>
> Although defining new objectives and models is not the main goal of this work, we agree with the reviewer that the combination of different criteria is an interesting direction for future research. This is true especially from a practical perspective since combining different estimators could help to reduce the bias/variance of the approximations.
> The relation between different criteria is also discussed in [1] and in appendix B.8.
>
> 3) **"How to numerically estimate the mutual information?"**
>
> To measure and optimize mutual information we use a discrete low-dimensional sufficient statistic ${\bf x}':=[c,d]$ (with $c$ being the color and $d$ as the digit) of the pictures ${\bf x}$ for which the joint density is known and compute it as in [2].
>
> We will clarify this in the main text, adding a detailed description in the appendix in which we show that ${\bf x}'$ is equivalent to ${\bf x}$ when it comes to estimating the quantities of interest.
>
>
> ## References:
>
> [1] *Barocas, Solon, Moritz Hardt, and Arvind Narayanan. "Fairness in machine learning." Nips tutorial 1 (2017): 2017.*
>
> [2] *Tishby, Naftali, and Noga Zaslavsky. "Deep learning and the information bottleneck principle." 2015 IEEE Information Theory Workshop (ITW). IEEE, 2015.*

---

### Decision · Program_Chairs · 2021-09-27

**Decision:**

Accept (Poster)

**Comment:**

The authors study out-of-distribution (OOD) generalisation using an information-theoretic framework, decomposing the contributions of concept and covariate shift to the overall error. They go on to show how the framework yields an upper bound on OOD error in the context of a (latent) representation of the input variables. Consistent with related results, this leads to a conclusion that to minimise OOD error necessitates losing some information which is predictive of the training data. The optimal tradeoff relies on data unavailable at training time and so the authors study 3 proxies for minimising concept shift in the representation: independence, sufficiency and separation. In an empirical evaluation, the authors study minimising these criteria in comparison with existing approaches with related objectives.

Several reviewers appreciated the information-theoretic formulation of OOD error and the theoretical treatment of it. Reviewer S3bF raised concerns about the significance of the technical contribution but this is to some extent compensated by the insights gained by the change in framework compared to related works. Additionally, several comments were made about the clarity of the organisation of the paper. In particular, transitions between sections are often hard to follow (see e.g., Section 2.2. which starts without any lead-in). The authors would do well to guide the reader better through the sections. Nevertheless, given the reviewers appreciation for the work, and the arguably minor concerns that remain after the discussion phase, I believe that a re-organisation of the paper is sufficient.